# Mutations in the *PIK3C2B*, *ERBB3*, *KIT*, and *MLH1* Genes and Their Relationship with Resistance to Temozolomide in Patients with High-Grade Gliomas

**DOI:** 10.3390/biomedicines12122777

**Published:** 2024-12-06

**Authors:** León Darío Ortiz Gómez, Heidy Johanna Contreras Martínez, David Andrés Galvis Pareja, Sara Vélez Gómez, Jorge Emilio Salazar Flórez, Fernando P. Monroy, Ronald Guillermo Peláez Sánchez

**Affiliations:** 1Doctoral Program in Health Sciences, Graduate School, CES University, Medellín 050021, Colombia; leonortizgomez@gmail.com; 2Cancer Institute, Las Americas-AUNA Clinic, Medellín 050023, Colombia; 3Pharmaceutical Sciences Research Group (ICIF), CES University, Medellín 050021, Colombia; hcontreras@ces.edu.co (H.J.C.M.); dgpareja@gmail.com (D.A.G.P.); 4Life and Health Sciences Research Group, Graduate School, CES University, Medellín 050021, Colombia; savelezg@uces.edu.co; 5GEINCRO Research Group, School of Health Sciences, San Martín University, Sabaneta 055457, Colombia; jorge.salazarf@sanmartin.edu.co; 6Department of Biological Sciences, Northern Arizona University, Flagstaff Arizona, AZ 85721, USA; fernando.monroy@nau.edu

**Keywords:** cancer, gliomas, chemotherapy, radiotherapy, temozolomide, resistance, mutations

## Abstract

Introduction. The treatment for patients with high-grade gliomas includes surgical resection of tumor, radiotherapy, and temozolomide chemotherapy. However, some patients do not respond to temozolomide due to a methylation reversal mechanism by the enzyme O^6^-methylguanine-DNA-methyltransferase (MGMT). In patients receiving treatment with temozolomide, this biomarker has been used as a prognostic factor. However, not all patients respond in the same way, which suggests the existence of other proteins involved in resistance to temozolomide chemotherapy. Methods. A group of thirty-one patients was recruited who were clinically and pathologically diagnosed with high-grade gliomas. The sequencing of 324 genes related to different types of cancer was performed to detect mutations. Subsequently, a statistical analysis was conducted to determine the mutated genes that were most related to resistance to treatment. Results. According to the Stupp protocol and metronomic dose of the temozolomide treatment, the mutated genes related to the second relapse of patients with high-grade glioma were *PIK3C2B*, *KIT*, *ERBB3*, and *MLH1*. Conclusions. Considering the results obtained, we suggest that mutations in the four genes and methylation of the gene promoter that codes for the MGMT protein could be related to response to treatment with temozolomide.

## 1. Introduction

According to the data provided by Cancer Today 2020, the incidence of brain tumors worldwide is 3.5 per 100,000 inhabitants, with a mortality rate of 2.8 per 100,000 inhabitants [1]. According to data obtained between 2003 and 2016, the incidence (standardized rate) of cancer in the brain and central nervous system in Colombia ranged from 3.5 to 4.2 per 100,000 inhabitants. While mortality ranged between 2.1 and 2.5 per 100,000 inhabitants between 1997 and 2020 (https://infocancer.co/, accessed October 25, 2024) [2], according to the Brain Tumor Registry (CBTRUS, 2014–2018), the incidence in the United States for primary central nervous system (CNS) tumors was 24 per 100,000, with a mortality rate of 4.3 per 100,000 [3]. Concerning the cases reported in the United States, these data reflect an underreporting of new cases in the rest of the world, including Colombia. Additionally, the CBTRUS estimates that for the United States, the incidence of gliomas is 5.95, and for the subgroup considered high-grade gliomas, glioblastoma is 3.23, anaplastic astrocytoma is 0.41, and anaplastic oligodendroglioma is 0.11 per 100,000 inhabitants [3]. The most abundant cells in the CNS are neurons and glial cells. The glial cells are composed of astrocytes, oligodendrocytes, ependymal cells, and microglia cells. Additionally, meningeal, and pituitary cells are also part of the CNS [4]. When cells have genetic and/or epigenetic alterations, mechanisms such as replication, repair, and senescence are modified, resulting in an uncontrolled proliferation of cells with aggressive characteristics that allow them to invade neighboring tissues and sometimes give rise to metastasis, even in isolated and immune privileged sites such as the brain parenchyma [5].

Tumors originating from glial cells are called gliomas. Glial cells are composed of four cell types: astrocytes, oligodendrocytes, ependymal cells, and microglia. The fifth edition of the World Health Organization’s classification of central nervous system tumors published in 2021 indicates that the classification of these tumors has changed, leaving behind terms such as grade 4 glioblastomas multiforme, oligoastrocytoma, and anaplastic astrocytoma [6]. The new classification recognizes 22 new tumor types and emphasizes the importance of histological and molecular diagnoses [7]. In the case of diffuse gliomas in adults, there are astrocytomas, which range from grade 2 to 4. It has been recognized that grade 4 IDH-mutated astrocytoma is described as a biologically distinct entity from glioblastoma [6]. Oligodendrogliomas are defined as gliomas with IDH mutation and 1p/19q codeletion, and they also usually have a mutation in the TERT promoter. Additionally, they are classified as grade 2 or 3 central nervous system tumors according to the WHO, depending on proliferation and anaplasia [8]. The most common IDH wild-type glioma is glioblastoma, grade 4. These tumors have an astrocytic morphology and show high-grade morphological features including necrosis and/or microvascular proliferation [6]. However, recent studies have demonstrated the existence of cellular heterogeneity within tumors, finding dynamic cell populations that may be related to patient survival [9].

High-grade gliomas, now named according to the 2021 WHO classification of tumors of the CNS, include isocitrate dehydrogenase (IDH) wild-type glioblastoma (GB^wt-IDH^), IDH-mutated astrocytoma without 1p/19q codeletion, grades 3 and 4 (A^mut-IDH,G3-4^), and oligodendroglioma mutated for IDH with codeletion 1p/19q grade 3 (O^mut-IDH,codel-1p/19q,G3^) [10,11]. The standard treatment for high-grade glioma is the protocol developed by Stupp et al. (2005) [12]. This protocol added temozolomide to radiotherapy treatment, reporting an increase in patient survival (HR not adjusted: 0.63, 95% CI: 0.52–0.75, 62 *p* < 0.001) and decreased the probability of death (HR: 0.51, 95% CI: 0.31–0.84) in patients in whom temozolomide was added to radiotherapy. These observations did not occur in patients who did not have the gene promoter encoding the MGMT protein methylated (HR: 0.69, 95% CI: 68 0.47–1.02) [13,14]. However, these findings indicate that this mechanism of resistance to temozolomide does not explain the response to treatment in 100% of the patients. Accordingly, there is a gap in our knowledge about other mechanisms or proteins involved in the process of resistance to treatment with temozolomide in patients with high-grade gliomas.

Therefore, the objective of our investigation was to identify additional genetic alterations to the promoter’s methylation of the gene that codes for the MGMT protein, which will help explain the response to treatment with temozolomide in patients with high-grade gliomas.

## 2. Materials and Methods

### 2.1. Study Population

The study population consisted of oncological patients who attended the Cancer Institute (Las Américas-AUNA Clinic, Medellín-Colombia) for management with surgery, radiotherapy, chemotherapy, and oncological support. The study was conducted by open invitation, recruiting patients who met the following inclusion criteria: patients over 18 years of age, diagnosed with a primary malignant brain tumor, who underwent tumor resection surgery or biopsy, and who should have a histological diagnosis of glioblastoma. A total of 31 patients diagnosed with high-grade glioma were included. According to the 2021 WHO classification of tumors of the central nervous system, the patients were classified as twenty-two glioblastomas, five astrocytomas, and four oligodendrogliomas. This study is classified as observational, cohort, and ambispective.

### 2.2. Clinical Information

The following data were extracted from clinical history: age, sex, clinical diagnosis, imaging diagnosis, pathology results, molecular biology tests, treatment, and outcomes.

### 2.3. Sample Collection for Molecular Biology Analysis

Total surgical resection was performed on all 31 patients to remove cancerous brain tissue. Twenty-six samples of cancerous brain tissue embedded in paraffin blocks and 5 samples of liquid biopsies (patients without paraffin blocks) were used to make DNA extraction, new generation sequencing, and genetic profiling of the 31 patients.

### 2.4. Sequencing 324 Cancer-Associated Genes

Genomic sequencing was performed using the FoundationOne^®^CDx (F1CDx) and FoundationONE Liquid CDx (F1LCDx) panels, which are an in vitro diagnostic methodology based on next-generation sequencing for the detection of mutations such as substitutions, insertions, and copy number alterations (CNAs) in 324 genes. Additionally, gene rearrangements and genomic fingerprinting, including microsatellite instability (MSI), tumor mutational burden (TMB), and loss of heterozygosity (LOH), are detected (https://www.foundationmedicine.com/test/foundationone-cdx, Cambridge, MA, USA, and https://www.foundationmedicine.com/test/foundationone-liquid-cdx, Cambridge, MA, USA; accessed on 25 October 2024).

### 2.5. Mutated Genes Identification Related to Temozolomide Resistance Mechanisms

The mutated genes identification related to temozolomide resistance mechanisms was carried out using the following procedures: A univariate analysis was performed, in which summary measures, including measures of central tendency, were calculated for quantitative variables, along with their respective measures of dispersion, according to the distribution of the variable (Shapiro–Wilk test). For qualitative variables, absolute and relative frequencies were calculated. The results were presented through graphs and tables. To perform the bivariate analysis, the Logrank test, simple Cox regression models, Kaplan–Meier analysis, and survival curves were used. The results were presented in tables and a survival curve graph. To perform the multivariate model, a multiple Cox regression model was performed. For the entry of the variables into the final model, the behavior of the bivariate analysis was considered; that is, those variables that met the following criteria were entered into the model: statistical significance with *p* < 0.05, Hosmer Lemechow criterion with *p* < 0.25, and according to the criteria of the investigator (biological plausibility). Variables were entered progressively, and interaction analysis was performed for each model until reaching the final model, considering the principle of parsimony. Statistical analyses were performed using STATA (version 14) and SPSS (version 28) statistical programs.

### 2.6. Bioinformatic Analysis of Mutations Effect on the Proteins

The chemical properties of the four proteins were identified using the ProtParam tool web server (https://web.expasy.org/protparam/, accessed on 6 September 2024). The identification of the chemical properties of the amino acids was conducted by means of the GPMAW web server (https://www.alphalyse.com/customer-support/gpmaw-lite-bioinformatics-tool/start-gpmaw-lite/, accessed on 6 September 2024) and the color protein sequence web server (https://npsa-prabi.ibcp.fr/cgi-bin/npsa_automat.pl?page=npsa_color.html, accessed on 6 September 2024). The signal peptide of the proteins was identified using the SignalP-6 web server (https://dtu.biolib.com/SignalP-6, accessed on 6 September 2024). The InterProScan web server was used to identify the domains, biological processes, molecular function, and cellular components of the four proteins (https://www.ebi.ac.uk/interpro, accessed on 6 September 2024). The three-dimensional structure of wild-type and mutated proteins was modeled using the SWISS-MODEL web server (https://swissmodel.expasy.org, accessed on 6 September 2024), and AlphaFold Server (https://alphafold.ebi.ac.uk/, accessed on 6 September 2024), and the structural change caused by the mutations was determined using the Swiss-pdb-viewer (Version 4.1) bioinformatics program (http://spdbv.unil.ch), to establish how the structure and functional domains of the protein were affected. Additionally, all the diseases that have been associated with mutations in the four genes were identified using the OMIM database (https://www.omim.org, accessed on 6 September 2024).

### 2.7. Prediction of Pathogenic SNPs

To predict the changes in the structure and functional domains of the protein produced by the SNPs, we used the SIFT (Sorting Intolerant From Tolerant) software (https://sift.bii.a-star.edu.sg/, accessed on 6 September 2024) and PolyPhen-2 (prediction of functional effects of human nsSNPs (http://genetics.bwh.harvard.edu/pph2/, accessed on 6 September 2024). PolyPhen-2 stratifies the SNPs into likely damaging, probably damaging, or benign, and generates a score between 0 and 1, where values (≤0.85) mean a greater probability of damage. SIFT generates two categories (tolerated/neutral) where values (≤0.05) are classified as damaging.

### 2.8. Prediction of Deleterious SNPs for the Protein

We used the PhD-SNP (Predictor of human Deleterious Single Nucleotide Polymorphisms) (https://snps.biofold.org/phd-snp/phd-snp.html, 6 September 2024) and SNPs and GO (predicting disease-associated variations using GO terms) (https://snps.biofold.org/snps-and-go/snps-and-go.html, accessed on 6 September 2024) web servers. Both predictors determine whether SNPs can cause disease. PhD-SNP classifies nsSNPs of a gene into neutral or human disease-causing mutations, while SNPs and GO predict disease-associated variations using GO terms; the harmful nonsynonymous SNPs predicted were analyzed with these two tools to determine their association with high-grade gliomas, and these were classified as diseases or neutral according to their potential to cause disease. SNPs labeled as a disease were retained for further analysis. The reliability index (RI) in PhD-SNP, and SNPs and GO ranges from 0 to 10, where 10 means the highest reliability.

### 2.9. Predicting Protein Stability for Functionally Deleterious SNPs

The SNPs were analyzed to determine their effects on the protein stability with the bioinformatics programs I-Mutant 3.0 (https://folding.biofold.org/i-mutant/i-mutant2.0.html, accessed on 6 September 2024), and MUpro (https://mupro.proteomics.ics.uci.edu/, accessed on 6 September 2024); the first one is an algorithm based on an SVM (Support Vector Machine), which automatically predicts the change in the stability of the proteins after a single nucleotide mutation, while the IMutant 3.0 algorithm was trained with a data set derived from Protherm that provides an estimate of the change in the Gibbs free energy (Delta Delta G or DDG), calculated by subtracting the DDG value of the non-mutated protein from the DDG value of the mutated protein (unit kcal/moles). Having a DDG > 0 means higher protein stability, while a DDG < 0 refers to lower protein stability. The MUpro server not only calculates using an SVM but also uses a neural network. It gives a prediction value between −1 and 1, where a value <0 indicates a decrease in stability.

### 2.10. Prediction of Protein Structural Alteration and Loss of Activity

To predict structural changes in the four proteins, including alteration in activity and binding, the MutPred2 web application (http://mutpred.mutdb.org/, accessed on 6 September 2024) was used, which is a method and software package based on machine learning that integrates genetic and molecular data to probabilistically reason about the pathogenicity of amino acid substitutions. It is trained on a set of 53,180 pathogenic and 206,946 unlabeled variants obtained from the Human Gene Mutation Database (HGMD), SwissVar, dbSNP, and pairwise alignment between species that can predict more than 50 protein features. The application provides different probability percentages for each of the characteristics; this probability varies between 0 and 1, where the score closest to 1 indicates that it is more likely that the mutation alters the property of the protein, generating a gain or loss of the function.

### 2.11. Structural Comparison Between Normal and Mutated Residues

The TM-Align application (https://zhanggroup.org/TM-align/, accessed on 6 September 2024) is an algorithm for sequence-independent protein structure comparisons. It gives a score between 0 and 1, where 1 indicates a perfect match between the mutated and wildtype protein structures, while a score ≥ 0.5 indicates that the same fold/topology exists between the structures located in the SCOP/CATH databases, which divides the protein into domains and classifies them at the hierarchical level. For example, SCOP classifies domains into classes, folds, superfamilies, and families. For its part, the four levels in CATH are class, architecture, topology, and homologous superfamily, thus generating a comparison between the structures. Scores of 0.2 correspond to unrelated proteins chosen at random. This tool was used to perform an analysis to establish a structural difference and thus conclude that SNPs affect the protein. Root means square deviation or RMSD scores were also provided, which measures the average distance between atoms of overlapping molecules, where higher scores indicate more structural differences between the two compared molecules and scores closer to 0 indicate greater geometric similarity between the two residues. The SWISS-MODEL program (https://swissmodel.expasy.org/, accessed on 6 September 2024) models the three-dimensional structure of proteins and provides the Ramachandran plot score between two structures. The two values, both the TM and the Ramachandran graph score, helped determine the structural differences caused by the alterations.

### 2.12. Group of Mutations

Mutation3D software (http://mutation3d.org/, accessed on 6 September 2024) was used, which predicts and visualizes the spatial arrangement of mutated amino acids in the protein structure, showing whether they are found in an important domain. It can also describe clusters of mutations to discover some more significant SNPs.

### 2.13. Observation of Amino Acid Change

The Project-HOPE database (https://www3.cmbi.umcn.nl/hope/, accessed on 6 September 2024) provides information on amino acid changes in protein structures along with changes in different physiological and chemical activities that occur after mutation. From this interface, it is possible to report whether the residue is in a conserved site, interpret mutations that are most likely to be harmful, and therefore significantly affect the function of the protein.

### 2.14. Protein–Protein Interaction

The protein interaction was studied using the STRING web server (https://string-db.org, accessed on 6 September 2024), which predicts the main proteins that show interactions with the query gene. STRING predicts such interactions based on gene fusion, co-expression, function, and experimental data. It shows combined scores for each interacting protein, ranging from 0 to 1, where 0 shows the lowest interaction and 1 indicates the highest interaction. In addition, the KEEG database (https://www.genome.jp/kegg/, accessed on 6 September 2024), and GeneCards database (https://www.genecards.org/, accessed on 6 September 2024) were used to identify the metabolic pathways.

## 3. Results

### 3.1. Study Population

A total of 31 patients met the inclusion criteria and agreed to participate in the study by signing the informed consent. In relation to the clinical data, the mean age at diagnosis of high-grade glioma was 47 years old (SD 14.52). Regarding gender, 11 patients were male (35.49%) and 20 were female (64.51%).

### 3.2. Clinical Information

The Karnofsky scale was applied to the 31 patients included in the study to assess their functional status at the time of diagnosis, finding values equal to or greater than 70% in all patients, which indicated that all were fit to undergo surgery. All patients were subjected to contrasted preoperative magnetic resonance imaging of the brain, which was recorded in the patient’s medical records, without discriminating by which cerebral hemisphere was injured and considering that sometimes the lesions are not confined to a single lobe. It was found that 32% of the patients had lesions in the frontal lobe, 16% in the parietal, 22% in the temporal, 26% in the occipital, and only one patient had a lesion in the basal ganglia. A grossly complete resection was performed on 45% of the patients, and a partial resection or biopsy was performed on the remainder (55%). According to the pathology report, based on hematoxylin-eosin (H-E) and immunohistochemistry (IHC), it was found that twenty-two (71%) of the patients had glioblastomas, five (16%) had astrocytoma, and four (13%) had oligodendroglioma (Figure 1). All patients were given concomitant treatment with temozolomide and radiotherapy, followed by adjuvant chemotherapy four to six weeks after surgery. In 15 patients, it was possible to prevent tumor growth with six cycles of temozolomide as recommended by the Stupp Protocol (SP). However, five patients received fewer cycles due to clinical deterioration and tumor growth (progression). In the remaining 11 patients, it was necessary to administer more than six cycles of temozolomide to prevent tumor growth (Figure 2 and Figure 3).

### 3.3. Sample Collection for Molecular Biology Analysis

To classify high-grade gliomas, the following molecular analyses were performed on the samples obtained from the first surgery: a search of the 1p/19q codeletion by FISH in three patients (10% of the sample), which helped to classify them as anaplastic oligoastrocytoma. Immunohistochemistry (IHC) for IDH1 was performed on 19 patients (61%), reporting positive results in 4/10 glioblastomas (40%), 3/4 anaplastic astrocytoma (75%), and in 5/5 oligodendroglial tumors (100%). The methylation status of the promoter of the gene that codes for the *MGMT* protein was searched by polymerase chain reaction (PCR) in 18 patients, finding methylation in 3/10 glioblastomas (30%), 2/3 anaplastic astrocytoma (66%), and 4/4 tumors of the oligodendroglial lineage (100%) (Appendix A).

### 3.4. Sequencing 324 Cancer-Associated Genes

The genetic material was extracted from the 31 samples, and 324 cancer-related genes were sequenced and analyzed according to the protocol described by the FoundationOne^®^CDx (F1CDx) and FoundationONE Liquid CDx (F1LCDx) panels. An amplification, sequencing, and mutation detection was conducted on each of the genes. According to the results obtained in the gene profiling of the 31 patients with high-grade gliomas, 370 types of mutations were found in 185 genes. The ten most frequently mutated genes, in order of frequency, for the 21 patients initially diagnosed with glioblastomas were *EGFR* (62%), *TERT* (62%), *PDGFA* (57%), *TP53* 43%, *CDKN2A/B* (29%), *NF1* (24%), *ALK* (19%), *EP300* (19%), *KDR* (19%), and *MLL2* (19%). The most frequently mutated genes in the five patients diagnosed with anaplastic astrocytoma were *IDH1* (40%), *TP53* (40%), *CCN2* (40%), *KEAP1* (40%), *KIT* (40%), *MAF* (40%), *PDGFRA* (40%), *SPEN* (40%), *TERT* (40%), and *TYRO3* (40%). The most frequently mutated genes in the group of five patients with anaplastic oligoastrocytomas were *TSC1* (75%), *SPEN* (50%), *IDH* (25%), *ALK* (25%), *ARID1A* (25%), *CIC* (25%), *MAP3K1* (25%), *MLL2* (25%), *NF1* (25%), and *POLE* (25%); See Appendix A.

### 3.5. Mutated Genes Identification Related to Temozolomide Resistance Mechanisms

The simple Cox’s regression analysis shows the comparison between the dependent variable “second relapse” (at 22 months) and the presence of mutated genes. Seventy-four mutated genes were found in patients who presented a second relapse. The genes with the greatest statistical significance were *PIK3C2B* HR 13.81 (95%, CI: 2.25–84.45, *p* = 0.004), *NOTCH3* HR 6.02 (95%, CI: 1.74–20.84, 199 *p* = 0.005), *KIT* HR 3.98 (95%, CI: 1.20–13.18, *p* = 0.024), *ERBB3* HR 3.87 (95%, CI: 1.06–14.04, 200 *p* = 0.04), and *MLH1* HR 3.52 (95%, CI: 0.95–13.09, *p* = 0.06); see Table 1.

As a second analysis, a multivariate analysis was performed to determine which mutated genes were related to the second relapse, reducing the number to four genes: *PIK3C2B* with an HR of 82.37 (95%, 203 CI: 8.36–111.67, *p* = 0.000), *KIT* with an HR 10.24 (95%, CI: 2.42–43.34, *p* = 0.002), *ERBB3* with an HR 13.20 (95%, CI: 2.77–62.77, *p* = 0.001), and *MLH1* with an HR 8.50 (95%, CI: 1.83–39.45, 205 *p* = 0.006); See Table 2.

As a third analysis, a survival curve was performed using the Kaplan–Meier method in patients with mutations in the four genes (*PIK3C2B*, *ERBB3*, *KIT*, and *MLH1*). This analysis was performed to estimate the time it takes for patients to reach a second relapse after treatment with metronomic temozolomide compared to patients without mutations in these genes. When analyzing the results of the survival curves of patients who had mutations, it was observed that they had a second relapse in a shorter time compared to the patients who did not have mutations in these four genes, as can be seen in Figure 4.

### 3.6. Bioinformatic Analysis of Mutations Effect on the Protein Domains

An in-silico bioinformatics analysis using the InterProScan web server was performed on the *PIK3C2B*, *ERBB3*, *KIT*, and *MLH1* mutated genes that were found in the second relapse after the use of temozolomide according to Stupp protocol and the administration of a metronomic dose. As a first analysis, the functional domains in the four proteins encoded by mutated genes were detected, and the different mutations found affected them. We found that for the PIK3C2B protein, the domain (PI3K_Ras-bd_dom) was affected; for the ERBB3 protein, the domain (Rcp_L) was affected; for the KIT protein, no mutations were found affecting functional domains; and for the MLH1 protein, two domains were affected (DNA_mismatch_repair_N and DNA_mismatch_S5_2-like). Additionally, the biological processes in which the mutated proteins participate were identified: PIK3C2B (Phosphatidylinositol-mediated signaling, and Phosphatidylinositol phosphatase biosynthetic process), ERBB3 (membrane receptor protein tyrosine kinase signaling pathway, and protein phosphorylation), KIT (protein phosphorylation, transmembrane receptor protein tyrosine kinase signaling pathway, KIT signaling pathway, and Fc receptor signaling pathway), and MLH1 (DNA mismatch repair). Also, the molecular functions in which the mutated proteins participate were identified: PIK3C2B (phosphatidylinositol binding, and kinase activity), ERBB3 (protein kinase activity, protein tyrosine kinase activity, and ATP binding), KIT (ATP binding, transmembrane receptor, protein tyrosine kinase activity, protein kinase activity, cytokine binding, and protein tyrosine kinase activity), and MLH1 (mismatched DNA binding, ATP binding, ATP-dependent DNA damage, and ATP hydrolysis activity).

Subsequently, the three-dimensional structure of each of the four proteins was modeled, and the structural effects caused by each of the mutations were determined. The PIK3C2B protein is made up of 1634 amino acids, has a molecular weight of 184,739.79 Da, a theoretical isoelectric point of 6.87, 194 negatively charged residues (Asp + Glu), 189 positively charged residues (Arg + Lys), an atomic composition of 8246 carbon, 12875 hydrogen, 2281 nitrogen, 2414 oxygen, and 67 sulfur atoms, and an estimated half-life of 30 h (mammalian reticulocytes, in vitro). The instability index was 46.22, and therefore the protein is predicted to be unstable. The R458Q mutation was detected in the protein. The wild-type amino acid is an arginine (R) that is classified as basic, which is replaced by a glutamine (Q) that is hydrophilic. The signal peptide was not detected using the SignalP bioinformatics program (version 6.0). The effect of the mutation on the structure of the protein leads to the shortening of the radical, reducing the Van der Waals forces, which possibly affects the attraction between atoms, molecules, and surfaces. At the structural level, the mutation is located between the amino acids (457–464) that make up a beta sheet, generating an elongation of the beta sheet in the mutated protein. All these alterations can lead to instability and loss of functionality of the protein; see Figure 5.

The ERBB3 protein is made up of 1342 amino acids, has a molecular weight of 148,098.19 Da, a theoretical isoelectric point of 6.11, 154 negatively charged residues (Asp + Glu), 132 positively charged residues (Arg + Lys), an atomic composition of 6447 carbon, 10126 hydrogen, 1852 nitrogen, 259 1969 oxygen, and 94 sulfur atoms, and an estimated half-life of 30 h (mammalian reticulocytes, in 260 vitro. The instability index was 49.61; therefore, the protein is predicted to be unstable. The R164K mutation was detected in the protein. The wild-type amino acid is an arginine (R) that is classified as basic, which is replaced by a lysine (K) that is hydrophilic. The signal peptide was not detected using the SignalP bioinformatics program (version 6.0). At the structural level, an amino acid change is observed that leads to an alteration in the orientation of the radical group that alters the spatial distribution of Van der Waals forces, which possibly affects the attractions between atoms, molecules, and surfaces; see Figure 5.

The KIT protein is made up of 977 amino acids, has a molecular weight of 109,992.70 Da, a theoretical isoelectric point of 6.54, 111 negatively charged residues (Asp + Glu), 106 positively charged residues (Arg + Lys), an atomic composition of 4925 carbon, 7657 hydrogen, 1301n, 1460 oxygen, and 48 272 sulfur atoms, and an estimated half-life of 30 h (mammalian reticulocytes, in vitro). The instability index was 39.43; therefore, the protein is predicted to be stable. No signal peptide was detected using the SignalP bioinformatics program (version 6.0). This protein did not present a structural alteration, but it showed an increase in the number of copies of the *KIT* gene in the patient’s genome. This genetic phenomenon is common in cancer cells, which produce several copies of one or more genes in response to signals from other cells or the environment.

The MLH1 protein is made up of 756 amino acids, has a molecular weight of 84,600 Da, a theoretical isoelectric point of 5.51, 104 negatively charged residues (Asp + Glu), 83 positively charged residues (Arg + Lys), an atomic composition of 3740 carbon, 5947 hydrogen, 1017 nitrogen, 1165 oxygen, and 25 sulfur atoms, and an estimated half-life of 30 h (mammalian reticulocytes, in vitro). The instability index was 51.29; therefore, the protein is predicted to be unstable. The F261fs*31 mutation was detected in the protein. The signal peptide was not detected using the SignalP bioinformatics program (version 6.0). The deletion of a cytokine at position 783 of the MLH1 gene (783delC), leads to structural change in the MLH1 protein (p.Phe261fs*7). This sequence change creates a premature translational stop signal of seven amino acids after phenylalanine (Frameshift). It is expected to result in an absent or disrupted protein product (this could lead to the production of a truncated protein of 268 amino acids); see Figure 5.

The diseases associated with each of the four mutated genes were identified using the OMIM database (An Online Catalog of Human Genes and Genetic Disorders). Mutations in the gene that codes for the PIK3C2B protein are associated with the following diseases: high-grade glioma, Maffucci, and Hepatitis C. Mutations in the gene that codes for ERBB3 protein are associated with the following diseases: high-grade glioma, breast cancer, frontal myelination, enteric nervous system, and diabetes type 1. Mutations in the gene that codes for KIT protein are associated with the following diseases: high-grade glioma, piebaldism, gastrointestinal stromal tumor (GIST), mastocytosis, liver cell membrane autoantibody (LMA), melanoma, and germ cell tumors. Mutations in the gene that codes for MLH1 protein are associated with the following diseases: high-grade glioma, hereditary nonpolyposis colorectal cancer (Lynch), and Turcot syndrome type I.

### 3.7. Prediction of Pathogenic SNPs

The R458Q-PIK3C2B protein, according to the results of PolyPhen-2, is predicted to be probably damaging with a score of 0.979 (sensitivity: 0.76; specificity: 0.96). According to the results of SIFT, substitution at position 458 from R to Q is predicted to be tolerated with a score of 0.45. The R164K-ERRB3 protein, according to the results of PolyPhen-2, this mutation is predicted to be probably benign with a score of 0.017 (sensitivity: 0.95; specificity: 0.80). According to the results of SIFT substitution at position 164 from R to K, it is predicted to be tolerated with a score of 0.57; see Table 3.

### 3.8. Prediction of Deleterious SNPs for the Protein

The PhD-SNP, SNPs and GO web servers were used to predict the deleterious effect of SNPs on proteins. The R458Q mutation in the PIK3C2B protein was predicted to have a neutral effect. The R164K mutation in the ERRB3 protein was predicted to have a neutral effect; see Table 3.

### 3.9. Predicting Protein Stability for Functionally Deleterious SNPs

The R458Q mutation in the PIK3C2B protein had a DDG value < 0 (−1.0766127), predicting less protein stability, using the MUpro bioinformatics program. The R164K mutation in the ERRB3 protein had a DDG value < 0 (−1.6580712) predicting less protein stability, using the MUpro bioinformatics program. Similarly, the mutations R458Q-PIK3C2B (DDG = −1.08) and R164K-ERRB3 (DDG = −1.47) were predicted to decrease protein stability using the bioinformatics program I-Mutant (version 3.0); See Table 3.

### 3.10. Prediction of Protein Structural Alteration and Loss of Activity

We analyzed SNPs using the MutPred2 web server to calculate alterations in protein properties, and structural alterations and loss of activity generally obtain scores higher than (0.70). According to the analysis, we found that R458Q-PIK3C2B (Score: 0.259) and R164K-ERRB3 (Score: 0.128) may not affect or slightly affect the properties of proteins; See Table 3.

### 3.11. Structural Comparison Between Normal and Mutated Residues

We ran the TM-aling and SWISS-MODEL programs to analyze the changes between the structures of the wild-type and mutated proteins. The lower TM score (values close to zero), and the higher RMSD value (greater than zero), and Ramachandran Graphic Score (percentages less than 90%), indicate major differences between protein structures. The R458Q-PIK3C2B mutation obtained the following values: TM-Score (0.999), RMSD Value (0,00), and Ramachandran Graphic Score (89.33), which indicates that the mutation does not significantly affect the structure of the protein. The R164K-ERRB3 mutation obtained the following values: TM-Score (0.980), RMSD Value (2,26), and Ramachandran Graphic Score (98.18), which indicates that the mutation does not significantly affect the structure of the protein; See Table 3.

### 3.12. Mutations Group

The Mutation-3D bioinformatics program was used to identify the spatial location of mutated amino acids in the protein structure, the location of mutations within or outside functional domains, mutation clusters, and the identification of sites with high mutation frequencies (hot spots). The R458Q-PIK3C2B mutation obtained the following values: location at a site of low mutational frequency (blue circle), the domain (PI3K_Ras-bd_dom) was affected, and the structure of the amino acid radical is altered. The R164K-ERRB3 mutation obtained the following values: location at a site of low mutational frequency (blue circle), the domain (Rcp_L) was affected, and the structure of the amino acid radical was altered; See Figure 6.

### 3.13. Observation of Amino Acid Change

The R458Q-PIK3C2B mutation obtained the following values: The mutant residue is smaller than the wild-type residue; The wild-type residue charge was positive, the mutant residue charge is neutral; The charge of the wild-type residue is lost by this mutation, which can cause a loss of interactions with other molecules; The wild-type residue forms a salt bridge with Aspartic Acid at position 456. The difference in charge will disturb the ionic interaction made by the original, wild-type residue. The R164K-ERRB3 mutation obtained the following values: The wild-type and mutant amino acids differ in size; The mutant residue is smaller than the wild-type residue, and this will cause a possible loss of external interactions; The wild-type residue forms a hydrogen bond with Alanine at position 172. The size difference between the wild-type and mutant residue means that the new residue is not in the correct position to make the same hydrogen bond as the original wild-type residue did; The wild-type residue forms a salt bridge with Aspartic Acid at position 162, Aspartic Acid at position 169, and Aspartic Acid at position 171. Therefore, these salt bridges could be altered; See Figure 6.

### 3.14. Protein–Protein Interaction

Using the STRING web server, the top ten proteins that interact with the proteins PIK3C2B, ERBB3, KIT, and MLH1 were identified. The values of the analysis range from 0 to 1 where 0 shows the lowest interaction and 1 indicates the highest interaction. The scores between PIK3C2B and the top ten proteins were as follows: PIK3C2G (Score: 0.967), INPPL1 (Score: 0.955), PIK3R1 (Score: 0.9539), PIK3R2 (Score: 0.952), PIK3R3 (Score: 0.948), PIK3C2A (Score: 0.941), PTEN (Score: 0.941), INPP4A (Score: 0.935), INPP4B (Score: 0.935), and PIKFYVE (Score: 0.929). The scores between ERBB3 and the top ten proteins were as follows: EGF (Score: 0.999), ERBB2 (Score: 0.999), EGFR (Score: 0.999), NRG1 (Score: 0.999), ERBB4 (Score: 0.999), NRG2 (Score: 0.999), GRB2 (Score: 0.999), SHC1 (Score: 0.999), TGFA (Score: 0.998), PIK3R1 (Score: 0.998). The scores between KIT and the top ten proteins were as follows: KITLG (Score: 0.999), GRB2 (Score: 0.998), PIK3R1 (Score: 0.995), CLEC11A (Score: 0.993), PTPN11 (Score: 0.990), PIK3CA (Score: 0.986), NRAS (Score: 0.981), KRAS (Score: 0.976), CXCL12 (Score: 0.974), and EGF (Score: 0.973). The scores between MLH1 and the top ten proteins were as follows: MSH2 (Score: 0.999), MSH6 (Score: 0.999), BRIP1 (Score: 0.999), MSH3 (Score: 0.999), PMS2 (Score: 0.999), BLM (Score: 0.999), MLH3 (Score: 0.999), EXO1 (Score: 0.999), PMS1 (Score: 0.999), and ATM (Score: 0.997). Additionally, using the KEEG and GeneCards databases, the top five metabolic pathways in which these proteins participate were identified; See Figure 7.

**Table 3 biomedicines-12-02777-t003:** Bioinformatics characterization of the mutations effect R458Q-PIK3C2B and R164K-ERRB3 proteins related to the second relapse after using temozolomide according to the Stupp protocol and a metronomic dose in patients with high-grade glioma. The table shows the analysis of the two mutations with the bioinformatics programs PolyPhen2, SIFT, PhD SNP, SNP and GO, MUpro, I-Mutant 3.0, MuPred2, TM-aling, and SWISS-MODEL to predict how these mutations might be affecting the structure and functional domains of the two proteins.

Chromosome	Position	RS	Gene	Amino Acid Alteration	Classification According to ClinVar	PolyPhen2 Score	Prediction PolyPhen	SIFT Score	Prediction SIFT	PhD SNP	SNP and GO
1	-	-	PIK3C2B	p.Arg485Gln	New	0.979	Probably damaging	0.45	Tolerated	Neutral	Neutral
12	56086452	rs560422339	ERBB3	p.Arg164Lys	Likely benign	0.017	Benign	0.57	Tolerated	Neutral	Neutral
**MUpro**								**I-Mutant 3.0**			
**Gene**		**Alteration**	**Value DDG** **(Kcal/mol)**		**Protein stability**	**Value DDG** **(Kcal/mol)**		**Reliability index**		**Protein stability**	
PIK3C2B		R458Q	−1.0766127		Decrease	−1.08		9		Decrease	
ERBB3		R164K	−1.6580712		Decrease	−1.47		7		Decrease	
		**MuPred2**									
**ID**		**Substitution**	**MuPred2 Score**			**Remarks**		**Affected PROSITE and ELM motifs**			
ERBB2		R164K	0.128			-		-			
PIK3C2B		R458Q	0.259			-		-			
**Gene**		**nsSNP**	**TM-aling**					**SWISS-MODEL**			
			**TM-score**			**Value RMSD**		**Ramachandran graphic score**			
PIK3C2B		R458Q	0.999			0.00		89.33%			
ERBB3		R164K	0.980			2.26		98.18%			

## 4. Discussion

According to the Central Brain Tumor Registry of the United States (CBTRUS), gliomas represent 25% of primary central nervous system tumors in adults and are more prevalent in men than in women. Oligodendrogliomas and astrocytoma predominate in the age range between 15 and 39 years. For those over 40 years of age, glioblastomas are the most frequent [4]. These results are consistent with our patient population, in which 21/31 patients presented with glioblastomas. The 31 patients included in this study had a mean age of 47 years (Standard Deviation (SD): 14.5), and women predominated (65.4%). This result is due to the small number of patients, which is not a representative sample of the population.

The patients with glial tumors were classified using conventional tests (codeletion 1p/19q (FISH), Mutation IDH (IHQ), MGMT promoter methylation (PCR)) [15], and genomic sequencing using the FoundationOne^®^CDx (F1CDx) and FoundationONE Liquid CDx (F1LCDx) panels. Currently, the diagnosis of high-grade gliomas includes the search for genetic mutations, epigenetic changes, and chromosomal alterations [16]. Additionally, IHC makes it possible to verify, among other things, if tumor cells are derived from glia [17,18]. Based on these analyses, we obtained the following results in the 31 patients with high-grade gliomas: twenty-two glioblastomas, five astrocytoma, and four oligodendroglioma.

Molecular biology studies are the cornerstones of the new 2021 WHO classification of tumors of the CNS and allow the classification of high-grade gliomas and some frequently associated mutations: glioblastoma (GB^wt-IDH^), in which mutations in the TERT promoter, EGFR promoter, chromosome 7 trisomy, and monosomy chromosome 10 are frequent; astrocytoma (A^mut-IDH,G3-4^), in which mutations in the ATRX, TP53 and CDKN2A/B genes can be found; and oligodendroglioma (O^mut-IDH,codel-1p/19q,G3^), in which TERT, CIC, FUBP1, and NOTCH1 promoter mutations are common (8, 18). In our study, patients with high-grade glioma had the following mutational frequencies according to the tumor variety: glioblastoma (GB^wt-IDH^), in which mutations in the TERT promoter (13/22 Patients), and EGFR promoter (5/22 Patients). Astrocytoma (A^mut-IDH,G3-4^), in which mutations in the ATRX (3/5 Patients), TP53 (4/5 Patients) and CDKN2A/B (1/5 Patients). Oligodendroglioma (O^mut-IDH,codel-1p/19q,G3^), in which TERT (3/4 Patients), CIC (3/4 Patients), FUBP1 (1/4 Patients), and NOTCH1 promoter (1/4 Patients), which corroborates the frequency of mutations in these genes in patients with high-grade gliomas.

Additionally, if an accurate diagnosis is made, we could predict life expectancy with greater certainty. In our group of patients, at the cut-off point of the study (two years), those patients with glioblastoma had a median survival of 29 months, not including two long-surviving patients. Only 2/5 of patients with astrocytoma died (one at 14.5 months and the other at 19 months), and all the patients with oligodendroglioma survived. Survival, based on the new 2021 WHO classification, has not yet been calculated for GB^wt-IDH^, A^mut-IDH,G3-4^, or O^mut-IDH,codel-1p/19q,G3^. Therefore, it could not be determined whether the survival of our patients was high or low.

The mechanism of action of temozolomide is to methylate the O^6^ position of guanines, which results in a mismatch between guanines and cytosines (O^6^ guanine pairing with thymine), which leads to activation of the MMR system [19,20,21]. This only repairs the chain that contains thymine, accumulating the chains with methylated guanines [22]. Therefore, intra and inter-covalent bonds occur, forming hairpins that prevent cell replication and lead to apoptosis [23]. However, cancer cells have a mechanism to reverse the effect of temozolomide through the transcription and subsequent translation of the MGMT protein, which, through a sulfhydryl group, removes the methyl group from O^6^ methyl-guanine, restoring guanine to its original form [24]. This is reflected when analyzing how an increase in survival was observed (HR: 0.51, 95%, CI: 0.31–0.84) in patients whose tumor had the *MGMT* gene promoter methylated, and who received radiotherapy and temozolomide, compared to those who received radiotherapy only. The patients who did not have promoter methylation and who received radiotherapy and temozolomide did not have a significant survival increase (HR: 0.69, 365 95%, CI: 0.47–1.02), when compared with those who only received radiotherapy (13, 14). In practice, the search for methylation of the *MGMT* gene only has prognostic value and does not predict response to treatment [25], so all patients with high-grade gliomas receive temozolomide [26]. Theoretically, patients with MGMT promoter methylation should be expected to respond 100% to treatment, but only a 51% decrease in risk of death has been observed. Additionally, patients without MGMT promoter methylation would be expected to have no response to treatment, but a 31% decreased risk of death has been observed [13,14]. This highlights the need to find alterations in other proteins to improve the prognosis and treatment response of patients with high-grade gliomas. Promoter methylation was identified in 17/31 patients (10 glioblastomas, 3 astrocytoma, and 4 oligodendrogliomas). The methylation data obtained in this study were consistent with what has been reported in the literature regarding different types of high-grade gliomas since methylation of the MGMT promoter is found in 50% of glioblastomas, in 75% of astrocytomas, and in almost all oligodendrogliomas [27]. In our study, Overall Survival (OS) in patients with glioblastoma promoter methylation was 59.2 months, compared to 24.6 months for those without promoter methylation. Overall survival cannot be calculated in patients with astrocytoma and oligodendroglioma, because the project lasted 24 months, and 5/9 patients were still alive. Therefore, progression-free survival (PFS) was analyzed. In the present investigation, we found that PFS at first and second relapse was higher in patients with gliomas who had the methylated promoter of the gene that codes for MGMT protein compared to those who did not. These results are consistent with those reported in the literature [27]. However, new biomarkers are required, which, in association with the methylation of the MGMT promoter, make it possible to predict the response to temozolomide in patients with high-grade glioma.

Therefore, in the present study, we performed the detection and analysis of mutations on 324 cancer-related genes in a group of 31 patients with high-grade gliomas to detect the genes that are related to temozolomide resistance. We found 185 mutated genes with different mutations. As a first statistical analysis, a bivariate analysis was conducted (Cox’s regression models, Kaplan–Meyer analysis, and survival curves) to evaluate the relationship of each mutated gene with the second relapse after the use of temozolomide as part of the Stupp protocol and metronomic dose, finding 71 genes related to second relapse. However, only the first five genes had significant *p*-values and hazard ratios; see Table 1. As a second statistical analysis, a multivariate analysis was performed, and Cox’s regression models were used to determine the mutated genes that were related to the second relapse. The genes with statistically significant values were *PIK3C2B* with a crude HR of 13.81 (95%, CI: 2.25–84.45, *p* = 0.004), *KIT* with a HR of 3.98 (95%, 399 CI: 1.20–13.18, *p* = 0.024), *ERBB3* with a HR of 3.87 (95%, CI: 1.06–14.04, *p* = 0.04), and *MLH1* with a HR of 3.52 (95%, CI: 0.95–13.09, *p* = 0.06); see Table 2. Additionally, progression-free survival (PFS) was assessed at first and second relapse among patients with and without mutations in the *PIK3C2B*, *ERBB3*, *KIT*, and *MLH1* genes. At the first relapse, the patients who had the mutations in these genes presented a lower PFS (8.85 months) than those who lacked the mutations (27.55 months). Also, PFS was compared for the second relapse and patients with mutations in these four genes had lower PFS (19.3 months) than those without mutations (38.17 months). Moreover, it was observed that patients who had mutations in the *PIK3C2B* gene had the lowest survival-free progression. Therefore, presenting mutations in any of these four genes increases the probability of suffering a relapse; See Figure 4.

In this study, we found that *PIK3C2B*, *ERBB3*, *KIT*, and *MLH1* genes had different types of mutations. According to the results of structural analysis, the PIK3C2B-R458Q mutation leads to the shortening of the radical, reducing the Van der Waals forces, and at the secondary structure level, the mutation is located between the amino acids (457–464) that make up a beta-sheet, generating an elongation of the beta-sheet in the mutated protein. Moreover, ERBB3-R164K mutation leads to an alteration in the orientation of the radical group that alters the spatial distribution of Van der Waals forces. In addition, MLH1-F261fs*31 mutation leads to the appearance of a premature stop codon that leads to the loss of the final region of the mutated protein; see Figure 5. On the other hand, SNP-type mutations are R458Q-PIK3C2B and R164K-ERRB3. These mutations were predicted to be non-pathogenic; however, a decrease in protein stability, alterations in structure, and impairment of functional domains were predicted; see Table 3 and Figure 6. The above mutations and their structural and functional effects could affect the structure and functional domains, leading to the loss of interactions and interruption of some metabolic pathways that could be related to the gliomagenesis process and resistance to temozolomide; see Figure 7.

These four proteins have been well characterized and studied at an experimental level. The PIK3C2B protein is involved in the biosynthetic and signaling process of phosphatidylinositol phosphatase called PIK3C2B (phosphatidylinositol-4-phosphate 3-kinase C2), which is part of a family of enzymes capable of phosphorylating the hydroxyl group at the 3′ position of the inositol ring of molecules, collectively called phosphatidylinositol, which convert phosphatidylinositol 4,5-bisphosphate (PIP2) into phosphatidylinositol 3,4,5-triphosphate (PIP3), and then to phosphorylate AKT, among others [28,29,30]. The *ERBB3* gene encodes a member of the epidermal growth factor receptor (EGFR) family of receptor tyrosine kinases. This membrane-bound protein has a neuregulin binding domain but not an active kinase domain. Therefore, it can bind this ligand but not convey the signal into the cell through protein phosphorylation. However, it does form heterodimers with other EGF-receptor family members which do have kinase activity. Heterodimerization leads to the activation of pathways, which leads to cell proliferation or differentiation. Amplification of this gene and/or overexpression of its protein have been reported in numerous cancer types, including prostate, bladder, and breast tumors [31,32]. The *KIT* gene encodes a receptor tyrosine kinase. It was initially identified as a homolog of the feline sarcoma viral oncogene v-kit and is often referred to as proto-oncogene c-Kit. The canonical form of this glycosylated transmembrane protein has an N-terminal extracellular region with five immunoglobulin-like domains, a transmembrane region, and an intra-cellular tyrosine kinase domain at the C-terminus. Upon activation by its cytokine ligand, stem cell factor (SCF), this protein phosphorylates multiple intracellular proteins that play a role in the proliferation, differentiation, migration, and apoptosis of many cell types and thereby plays an important role in hematopoiesis, stem cell maintenance, gametogenesis, melanogenesis, and in mast cell development, migration, and function. This protein can be a membrane-bound or soluble protein [33,34]. The *MLH1* gene, the protein encoded by this gene can heterodimerize with mismatch repair endonuclease PMS2 to form MutL alpha, part of the DNA mismatch repair system. When MutL alpha is bound by MutS beta and some accessory proteins, the PMS2 subunit of MutL alpha introduces a single strand break near DNA mismatches, providing an entry point for exonuclease degradation. This protein is also involved in DNA damage signaling and can heterodimerize with DNA mismatch repair protein MLH3 to form MutL gamma, which is involved in meiosis [21].

According to the previous biological functions, three mutated proteins could be related to the gliomagenesis process and one of them to the mechanisms of resistance to temozolomide. Regarding to gliomagenesis process, the KIT gene, which encodes a class III receptor tyrosine kinase (RTK) [35,36], is frequently involved in tumorigenic processes. In the case of gliomas, it has been identified that genesis and progression can be driven in part by populations of cancer stem cells, which have self-renewal and differentiation potential [37]. It is known that some gliomas can arise from a cell with properties like those of neural stem cells [38]. In this case, KIT is expressed in human bone marrow progenitor cells and in rodent glial progenitor cells, but its expression is lost when glial cells differentiate into oligodendrocytes [39]. The stem cell factor is upregulated in high-grade human gliomas, promotes angiogenesis and its expression is associated with a short survival. Stem cell factor and KIT activation may play a role in the development and progression of glioma [13], and its mutation frequency is 4.4% in glioblastomas [40].

On the other hand, the PIK3C2B gene is in a member of the PIK3 family, which are lipid kinases involved in multiple cellular processes, including proliferation, differentiation, metabolism and survival [41]. This gene is involved in the PI3K/Akt/mTOR signaling pathway, which has been confirmed to play an important role in the genesis and development of glioma [42]. Glioblastoma patients with an activated PI3K/Akt/mTOR pathway also have a worse prognosis than patients without oncogenic activation of the pathway [42]. PIK3Ks transduce signals from various growth factors and cytokines into intracellular messages by generating phospholipids, which in turn activate the serine/threonine kinase AKT and other downstream signaling pathways, within these pathways is the tumor suppressor gene PTEN, which is the most important negative regulator of the PI3K/Akt/mTOR signaling pathway [43]. The mutation frequency of PIK3C2B in high-grade gliomas is unknown. On the other hand, the ERBB3 gene which belongs to the epidermal growth factor receptor (EGFR) family, which encompasses four members that are activated upon ligand-induced homodimerization or heterodimerization [31,32,44], this gene is associated with the regulation of normal cell proliferation, apoptosis, differentiation and survival [31,32,44]. When phosphorylated, ERBB3 shows a peculiar ability to maintain the pro-survival signaling of phosphatidylinositol 3-kinase (PI3K)/AKT, which may contribute to resistance against conventional and targeted therapies, and recurrence in a wide range of tumors. Therefore, ERBB3 itself is the subject of increasing interest as a therapeutic target [31,32]. Overexpression of ERBB3 has been shown to be associated with short survival of primary glioblastoma and is more frequent (up to 62%) in recurrent gliomas [31,32,44], suggesting that high ERBB3 activity may be involved with tumor aggressiveness and resistance to chemotherapy. Overexpression and activation of this gene sustains enhanced cellular metabolism, with increased extracellular acidification, oxidative phosphorylation and de novo fatty acid biosynthesis, which correlates with increased survival in vitro and tumorigenic potential in vivo [31,32,44].

Regarding to resistance mechanism, alterations in genes that are part of the DNA mismatch repair system have been reported in recurrent glioblastomas after the use of temozolomide [45]. Analysis of the Cancer Genome Atlas (TCGA) revealed a hypermutator phenotype, with mutations in at least one of the MMR genes (*MLH1*, *MSH2*, *MSH6*, or *PMS2*), suggesting either an escape from MGMT methylation or the selection of MMR mutated clones [46]. Felsberg et al. reported changes in promoter methylation and expression of the *MGMT*, *MLH1*, *MSH2*, *MSH6*, and *PMS2* genes after relapse in 80 patients with glioblastomas. They found that only four patients (6.25%) had a loss or decreased methylation of the MGMT promoter at recurrence. Although none of the four genes that are part of MMR had promoter hypermethylation, they did have mutations, which was confirmed by IHC [47]. DNA double-strand breaks by temozolomide activating the homologous and non-homologous repair mechanisms, which, through the ataxia telangiectasia (ATM)-checkpoint kinase 2 (CHK2) pathways, lead to p53 damage repair caused by temozolomide or trigger apoptosis [48]. Other mechanisms of resistance to temozolomide consist of the proper functioning or overactivation of some of the BER components, such as APGN, which would repair methylation at N7 of guanine and N3 of adenine [49]. Resistance to temozolomide can develop due to the overactivation of MDM2, which increases the X-linked inhibitor of the apoptosis protein (XIAP). This key regulator of both intrinsic and extrinsic programmed cell death signaling functions by suppressing the activation of caspases 3, 7, and 9, triggering their degradation mediated by ubiquitination or by the improper functioning of p53; therefore, it is unable to activate the Bcl2 family and the activation of the DR5 receptor [50]. Therefore, the MLH1 mutated gene could be involved in one of these resistance mechanisms.

Bioinformatics studies are of great importance because they allow us to organize information from large volumes of scientific data, analyze them, and obtain preliminary conclusions about the results of the research. These results can be verified experimentally to corroborate their validity; however, having bioinformatics results directs experimental studies to obtain better results. The sample size of 31 patients is very small for a genomic study. This limits the statistical power and generalizability of the findings. Therefore, we recognize that the sample size is the main limitation of the study.

## 5. Conclusions

We considered the early relapses of patients before six cycles of treatment with temozolomide, the decreased survival in patients with the four mutated genes, the structural alterations in the four proteins, the functional alterations caused by the mutations affecting the functional domains and decreasing the stability of the four mutated proteins, and the possible loss of interactions and interruption of metabolic pathways important for proper cellular functioning. We suggest that mutations in the four genes and methylation of the gene promoter that codes for the MGMT protein could be related to the response to temozolomide treatment. These results are exploratory and preliminary but will be the basis for a subsequent experimental study to verify the validity of these findings.

## Figures and Tables

**Figure 1 biomedicines-12-02777-f001:**
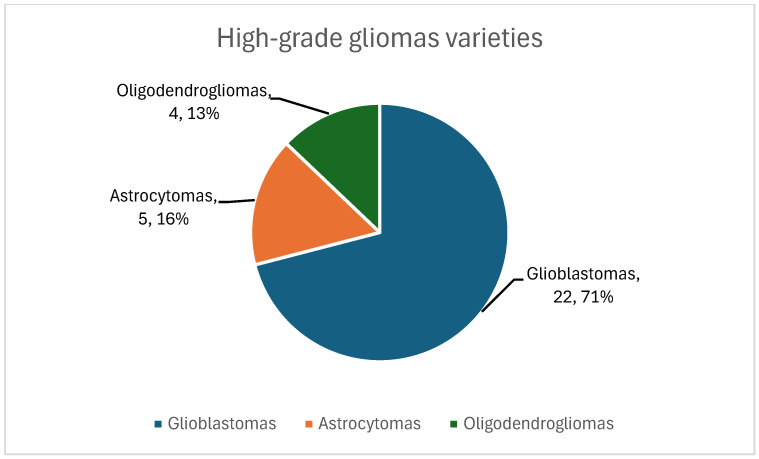
Type of tumor of patients with high-grade glioma. The figure shows the classification of the 31 patients analyzed according to the type of tumor: glioblastoma (22 patients; 71%), astrocytoma (5 patients; 16%), and oligodendrogliomas (4 patients; 13%). The most frequent tumor was glioblastoma.

**Figure 2 biomedicines-12-02777-f002:**
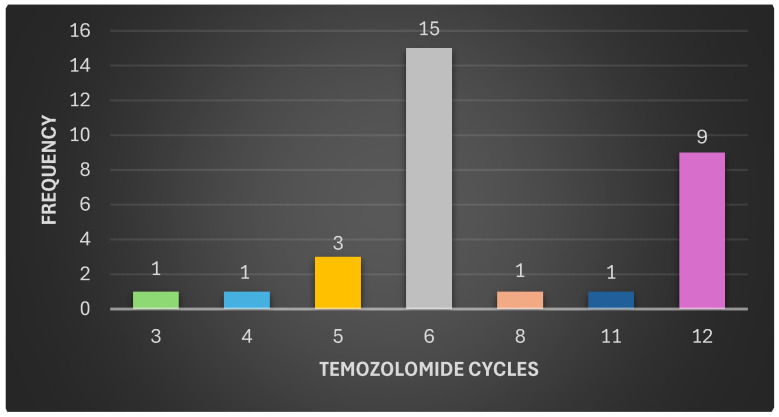
The figure shows the treatment cycles with temozolomide given to the 31 patients with high-grade glioma. Fifteen patients received the six cycles recommended by the Stupp protocol, five patients received fewer cycles due to an early relapse, and eleven patients needed more cycles due to disease persistence.

**Figure 3 biomedicines-12-02777-f003:**
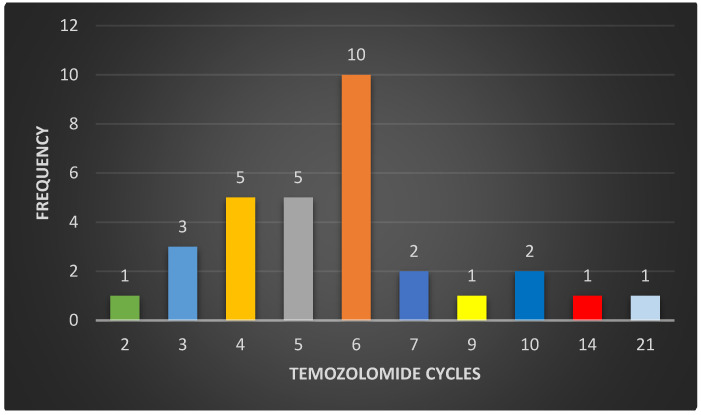
The figure shows the number of treatment cycles with metronomic temozolomide that patients received after the first relapse and until the second relapse occurs. The largest number of patients presented a second relapse at the sixth treatment cycle (10 patients), but the results were highly variable, with early (second cycle) and late (21 cycles) second relapses.

**Figure 4 biomedicines-12-02777-f004:**
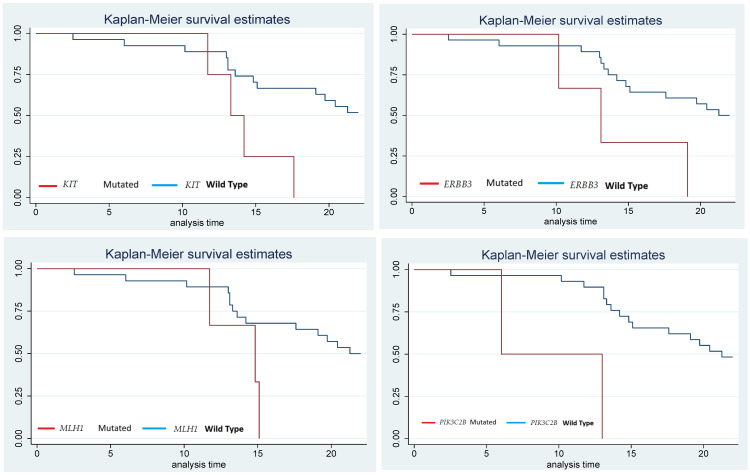
The figure shows the survival curves according to the Kaplan–Meier method. This analysis was performed to estimate the time it takes for patients with and without mutations in the *PIK3C2B*, *KIT*, *ERBB3*, and *MLH1* genes to present a second relapse after treatment with metronomic temozolomide. The graph shows patients without mutations with blue lines, while the red lines represent patients with mutations, observing that patients with mutations present a second relapse in less time. The Y-axis represents the survival rate, and the X-axis shows time in months, all patients were followed for 22 months.

**Figure 5 biomedicines-12-02777-f005:**
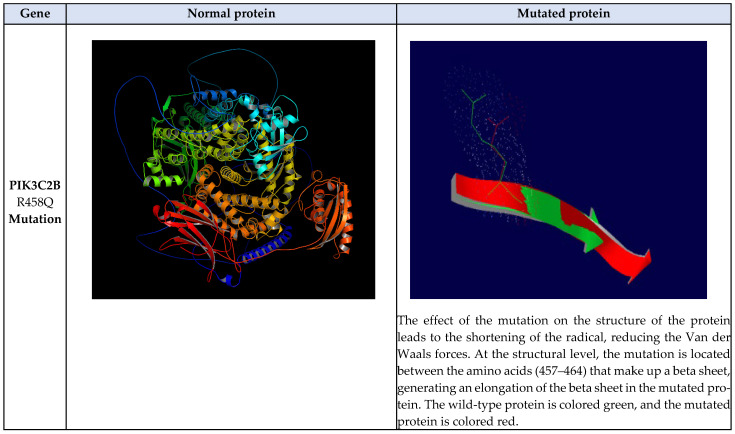
Structural analysis of the effect of the mutations in the four proteins encoded by mutated genes related to the second relapse after the use of temozolomide according to the Stupp protocol and a metronomic dose in patients with high-grade gliomas. The first column shows the protein name and mutation found. The second column shows the structural modeling of the wild-type protein. The third column shows the structural alignment and effect of the mutations on protein structure.

**Figure 6 biomedicines-12-02777-f006:**
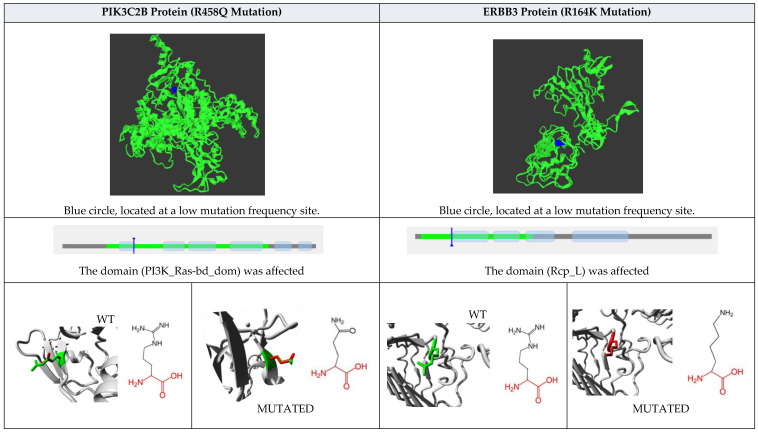
Bioinformatics analysis of the effect on structural and functional domains caused by mutations in the PIK3C2B (R458Q) and ERBB3 (R164K) proteins. The mutations were analyzed with the bioinformatics programs Project Hope and Mutation 3D. The first row describes the protein and its respective mutation. The second row shows the three-dimensional structure of the protein and the exact location of the mutation in the three-dimensional structure. The blue circle means that the mutation is located at a low mutation frequency site. The third row shows a linear image of the protein, the domains it possesses, and the exact location affected by the mutation. The fourth row magnifies the mutation site and structurally compares the wild-type and mutated amino acids (red color indicates the common structure between amino acids and the black structure the differences).

**Figure 7 biomedicines-12-02777-f007:**
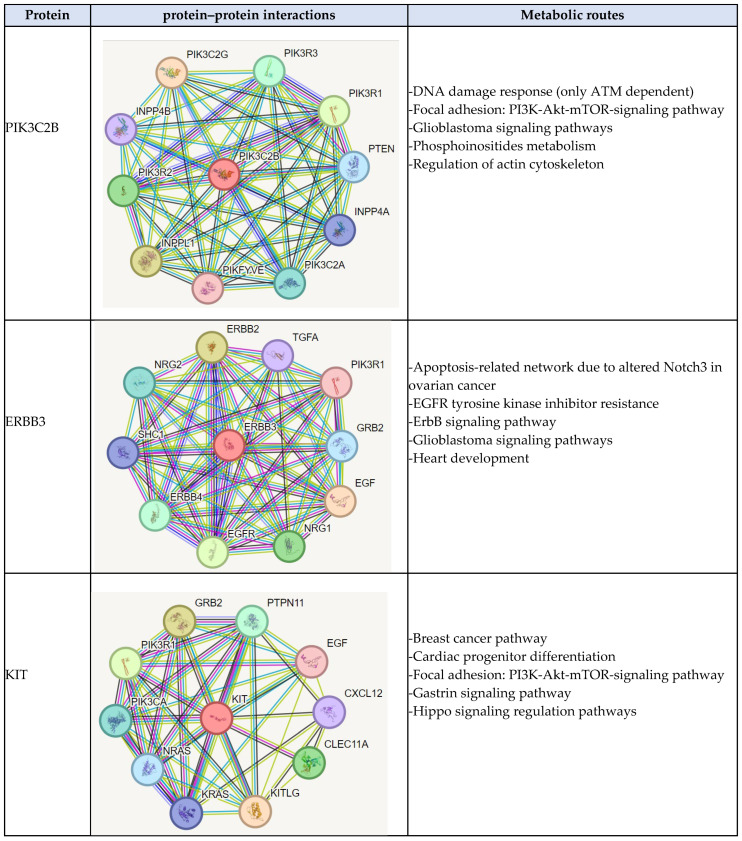
Identification of the protein–protein interactions and the metabolic pathways of the proteins encoded by the four mutated genes related to the second relapse after the use of temozolomide according to the Stupp protocol and the administration of metronomic doses in patients with high-grade glioma.

**Table 1 biomedicines-12-02777-t001:** Bivariate analysis was used to identify the mutated genes associated with the second relapse after the use of temozolomide according to the Stupp protocol and the metronomic dose in patients with high-grade gliomas. Seventy-one genes were related to the second relapse; however, only the first five genes with significant values of *p*-values and hazard ratios are shown.

Gene	*p*-Value	Hazard Ratio (CI 95%)
*PIK3C2B*	0.004	13.81 (2.258–84.45)
*NOTCH3*	0.005	6.026 (1.742–20.84)
*KIT*	0.024	3.985 (1.204–13.18)
*ERBB3*	0.04	3.871 (1.066–14.04)
*MLH1*	0.06	3.528 (0.950–13.09)

**Table 2 biomedicines-12-02777-t002:** Multivariate analysis was used to identify the mutated genes associated with the second relapse, after the use of temozolomide, according to the Stupp protocol and metronomic dose, in patients with high-grade gliomas.

Gene	*p*-Value	HR Raw Value (CI 95%)	*p*-Value	HR Adjusted Value (CI 95%)
*PIK3C2B*	0.004	13.81 (2.258–84.45)	0.0001	82.37 (8.36–811.67)
*KIT*	0.024	3.985 (1.204–13.18)	0.002	10.24 (2.42–43.34)
*ERBB3*	0.04	3.871 (1.066–14.04)	0.001	13.20 (2.77–62.77)
*MLH1*	0.06	3.528 (0.950–13.09)	0.006	8.50 (1.83–39.45)

## Data Availability

The mutation sequences were submitted to the NCBI database under the following accession numbers (PIK3C2B ID: 2874485, ERBB3 ID:2874518 and MLH1 ID:2874529).

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
