# Peer review of "Mutations in the PIK3C2B, ERBB3, KIT, and MLH1 Genes and Their Relationship with Resistance to Temozolomide in Patients with High-Grade Gliomas"

_biomedicines, 2024, doi:10.3390/biomedicines12122777_

Round 1
Reviewer 1 Report
Comments and Suggestions for Authors
The manuscript provides valuable analysis of the relationship of temozolomide resistance with a genetical pattern of high-grade glial tumours. The results are well presented. However, the discussion is written using an intricate language, thus it is hard to read, especially for general readers. It would be highly beneficial to rewrite the discussion, probably using subheadings and local summaries. Additionally, the Conclusions section is expected at the end of such a detailed discussion.
Here is the list of specific commentaries:
Lines 59-61. Glioblastomas surprisingly missed there, occurring only a few sentences later without a linkage to the cell from which they originated. Consider rewriting this section. Moreover, I suggest to connect this section to the recent findings in cell heterogeneity within a tumour as it can be relevant to the presented findings or highlight further directions (see doi: 10.3390/cancers15010145 and works of Cyril Neftel cited there).
Lines 97-99. Please check the section as it appears to be unclear.
Figures 1-3. Consider adding the information on the resection type (whether complete or partial) to the figures (highlighting respective parts of diagrams with alternative colours) and relative discussion.
Table 1. Please check the columns’ captions.
Line 229 and Table 2. Please check the correctness of presenting a zero p-value.
Figure 4. Consider redesigning a figure to make figure legend more intuitive.
Lines 253-255. What does it mean: “genes found to be related”? The sentence is ambiguous as causality is not determined. I do not think the authors meant that the genes appear to be mutated during the relapse. Consider rewriting the sentence.
Table 3. The reason for the addition of a table in such a design is not clear to me. Do the functions and processes describe the particular domains or the overall protein? Which processes and functions alter due to mutations and how, and which are unaffected? Is “none” in the Cellular Component mean? The protein was not located in a cell. Please clarify the table by correcting and expanding it, or remove the table and adding the necessary information to the main text.
Table 4. The quality of the figures is insufficient. Moreover, three of four proteins interact with the same proteins. Consider redesigning the figures by combining them to reflect all possible interactions (thus, the table should be modified or replaced).
Table 6 looks redundant after the text section containing all the presented information. Consider the elimination of the table.
Page 15. Please remove the excessive reference to Supplementary Material 1.
Comments on the Quality of English Language
Please, check the translation at figures and tables.
The language in the Discussion section requires the editing for fluency of the presentation.
Author Response
Comments and Suggestions for Authors
The manuscript provides valuable analysis of the relationship of temozolomide resistance with a genetical pattern of high-grade glial tumors. The results are well presented. However, the discussion is written using an intricate language, thus it is hard to read, especially for general readers. It would be highly beneficial to rewrite the discussion, probably using subheadings and local summaries.
The discussion was rewritten, using subtopics and local summaries.
Additionally, the Conclusions section is expected at the end of such a detailed discussion.
The conclusions section is added.
Here is the list of specific commentaries:
Lines 59-61. Glioblastomas surprisingly missed there, occurring only a few sentences later without a linkage to the cell from which they originated. Consider rewriting this section. Moreover, I suggest connecting this section to the recent findings in cell heterogeneity within a tumour as it can be relevant to the presented findings or highlight further directions (see doi: 10.3390/cancers15010145 and works of Cyril Neftel cited there).
The reviewer is correct:
Gliomas and the cells from which they originate are added. In addition, the recent discovery of cellular heterogeneity within tumors and its possible relationship to patient survival is described, citing the reference suggested by the reviewer.
Tumors originating from glial cells are called gliomas. Glial cells are composed of four types of cells: astrocytes, oligodendrocytes, ependymal cells, and microglia, this last cell is part of the immune system. Tumors arising from these cell types include as-trocytoma, oligodendrogliomas, and ependymomas, the latter of which are increasingly considered to be of a different strain. Astrocytoma is classified as low and high grade. Low-grade astrocytoma is grade 1, which are managed only surgically and include Ju-venile Pilocytic Astrocytoma (JPA), Pilomyxoid Astrocytoma (PMA), Fibrillary Astro-cytoma, Pleomorphic Xanthoastrocytoma (PXA), and Diffuse Astrocytoma. The grade 2 astrocytoma that belong to the family of diffuse infiltrating gliomas. High-grade as-trocytoma is grade 3 or anaplastic astrocytoma, and grade 4 or glioblastoma multiforme. Grade 2 oligodendroglioma is considered low grade, and grade 3 called anaplastic oli-godendroglioma is high grade. Recent studies have demonstrated the existence of cel-lular heterogeneity within tumors, finding dynamic cell populations that may be related to patient survival (6).
- Nikitin PV, Musina GR, Pekov SI, Kuzin AA, Popov IA, Belyaev AY, Kobyakov GL, Usachev DY, Nikolaev VN, Mikhailov VP. Cell-Population Dynamics in Diffuse Gliomas during Gliomagenesis and Its Impact on Patient Survival. Cancers (Basel). 2022 Dec 26;15(1):145. doi: 10.3390/cancers15010145. PMID: 36612140; PMCID: PMC9818344.
Lines 97-99. Please check the section as it appears to be unclear.
The evaluator is right, we correct the paragraph as follows:
Twenty-six samples of cancerous brain tissue embedded in paraffin blocks and 5 samples of liquid biopsies were used to make DNA extraction, new generation sequencing, and genetic profiling of the 31 patients.
Figures 1-3. Consider adding the information on the resection type (whether complete or partial) to the figures (highlighting respective parts of diagrams with alternative colours) and relative discussion.
Information on the type of surgical recession is added. The figures were recreated with alternating colors.
Total surgical resection was performed on all 31 patients to remove cancerous brain tissue. Twenty-six samples of cancerous brain tissue embedded in paraffin blocks and 5 samples of liquid biopsies (patients without paraffin blocks) were used to make DNA extraction, new generation sequencing, and genetic profiling of the 31 patients.
Table 1. Please check the columns’ captions.
The column headings in Table 1 have been adjusted.
Line 229 and Table 2. Please check the correctness of presenting a zero p-value.
The evaluator is right, one more decimal is added to the p-value, so that it does not equal zero.
Figure 4. Consider redesigning a figure to make figure legend more intuitive.
The quality of the figure is improved, and the description is improved.
Lines 253-255. What does it mean: “genes found to be related”? The sentence is ambiguous as causality is not determined. I do not think the authors meant that the genes appear to be mutated during the relapse. Consider rewriting the sentence.
The evaluator is right, the sentence is corrected as follows:
An in-silico bioinformatics analysis was performed on the PIK3C2B, ERBB3, KIT, and MLH1 mutated genes that were found in the second relapse after the use of temozolomide according to the Stupp protocol and the administration of a metronomic dose.
Table 3. The reason for the addition of a table in such a design is not clear to me. Do the functions and processes describe the particular domains or the overall protein? Which processes and functions alter due to mutations and how, and which are unaffected? Is “none” in the Cellular Component mean? The protein was not located in a cell. Please clarify the table by correcting and expanding it or remove the table and adding the necessary information to the main text.
The table was deleted, and the information was added to the text in the following paragraph:
An in-silico bioinformatics analysis was performed on the PIK3C2B, ERBB3, KIT, and MLH1 mutated genes that were found in the second relapse after the use of te-mozolomide according to the Stupp protocol and the administration of a metronomic dose. As a first analysis, the functional domains in the four proteins encoded by mutated genes were detected, and whether the different mutations found affected them. Finding that for the PIK3C2B protein, the domain (PI3K_Ras-bd_dom) was affected; for the ERBB3 protein, the domain (Rcp_L) was affected; for the KIT protein, no mutations were found affecting functional domains; and for the MLH1 protein, two domains were af-fected (DNA_mismatch_repair_N and DNA_mismatch_S5_2-like). Additionally, the biological processes in which the mutated proteins participate were identified: PIK3C2B (Phosphatidylinositol-mediated signaling, and Phosphatidylinositol phosphatase bio-synthetic process), ERBB3 (membrane receptor protein tyrosine kinase signaling pathway, and protein phosphorylation), KIT (protein phosphorylation, transmembrane receptor protein tyrosine kinase signaling pathway, KIT signaling pathway, and Fc re-ceptor signaling pathway), and MLH1 (DNA mismatch repair). Also, the molecular functions in which the mutated proteins participate were identified: PIK3C2B (phos-phatidylinositol binding, and kinase activity), ERBB3 (protein kinase activity, protein tyrosine kinase activity, and ATP binding), KIT (ATP binding, transmembrane receptor, protein tyrosine kinase activity, protein kinase activity, cytokine binding, and protein tyrosine kinase activity), and MLH1 (mismatched DNA binding, ATP binding, ATP-dependent DNA damage, and ATP hydrolysis activity).
Table 4. The quality of the figures is insufficient. Moreover, three of four proteins interact with the same proteins. Consider redesigning the figures by combining them to reflect all possible interactions (thus, the table should be modified or replaced).
The quality of the figure was improved, and the interaction network for each of the 4 mutated proteins was expanded.
Table 6 looks redundant after the text section containing all the presented information. Consider the elimination of the table.
The reviewer is right, Table 6 was removed from the text.
Page 15. Please remove the excessive reference to Supplementary Material 1.
The reviewer is right, excessive references to supplementary material 1 were removed and only the first reference was left.
Comments on the Quality of English Language
- Please, check the translation at figures and tables.
The translation of the figures and tables was corrected
- The language in the Discussion section requires the editing for fluency of the presentation.
The English language of the discussion was corrected to make it more fluid.
¡¡¡We greatly appreciate the evaluator's corrections, which help to greatly improve the research article!!!
Reviewer 2 Report
Comments and Suggestions for Authors
This study investigated mutations in the PIK3C2B, ERBB3, KIT, and MLH1 genes and their potential relationship with resistance to temozolomide treatment in patients with high-grade gliomas. The authors performed genetic profiling on tumor samples from 31 patients and analyzed associations between mutations and clinical outcomes. While the topic is relevant and interesting, there are several major concerns with the study design and presentation of results that need to be addressed:
Major comments:
- The sample size of 31 patients is very small for a genomic study. This limits the statistical power and generalizability of the findings. The authors should acknowledge this as a major limitation.
- The patient cohort is heterogeneous, including different types of high-grade gliomas. Combining these different tumor types in the analyses may confound the results. Separate analyses for each tumor type would be more appropriate.
- The statistical analyses and criteria for identifying genes of interest are not clearly described. More details are needed on the bioinformatics and statistical methods used.
- The functional significance of the identified mutations is not well characterized. Additional experiments would be needed to demonstrate a causal role in temozolomide resistance.
- The discussion section is overly long and speculative given the limited data presented. This should be considerably shortened and focused on the actual findings.
- The English writing needs substantial improvement throughout. There are numerous grammatical errors and awkward phrasings that make the manuscript difficult to follow in places.
Specific comments:
- The introduction lacks a clear rationale for focusing on these particular 4 genes. Why were these selected for analysis out of the many genes sequenced?
- Figure 4 is of poor quality and difficult to interpret. This should be redrawn for clarity.
- The bioinformatics analyses in section 3.6 are not well integrated with the clinical data. How do these predicted functional effects relate to the observed clinical outcomes?
- References need to be formatted consistently according to journal guidelines.
Comments on the Quality of English Language
The manuscript contains numerous grammatical errors that significantly impair readability. Here are some examples of issues that need to be addressed:
Subject-verb agreement error: "Its instability index is computed to be 46.22 and classified as unstable."
->"Its instability index is computed to be 46.22 and it is classified as unstable."
Incorrect preposition use: "No signal peptide was detected using the SignalP bioinformatics program (version 6.0)."
->"No signal peptide was detected by the SignalP bioinformatics program (version 6.0)."
Author Response
This study investigated mutations in the PIK3C2B, ERBB3, KIT, and MLH1 genes and their potential relationship with resistance to temozolomide treatment in patients with high-grade gliomas. The authors performed genetic profiling on tumor samples from 31 patients and analyzed associations between mutations and clinical outcomes. While the topic is relevant and interesting, there are several major concerns with the study design and presentation of results that need to be addressed:
Major comments:
- The sample size of 31 patients is very small for a genomic study. This limits the statistical power and generalizability of the findings. The authors should acknowledge this as a major limitation.
The reviewer is right, you have attached the following paragraph in the materials and methods section and the study population subsection:
The sample size of 31 patients is very small for a genomic study. This limits the statistical power and generalizability of the findings. Therefore, we recognize that the sample size is the main limitation of the study.
Additionally, in Colombia, the amounts funded for research are not high. Therefore, we analyzed only 31 patients using next-generation sequencing. However, this study will serve as a basis for larger studies to be carried out, with greater sources of funding. For now, the results are limited to the population of patients analyzed.
- The patient cohort is heterogeneous, including different types of high-grade gliomas. Combining these different tumor types in the analyses may confound the results. Separate analyses for each tumor type would be more appropriate.
The reviewer is right, but the decision to analyze the 31 patients with high-grade gliomas is because the sample is very small.
If we subdivide the sample into:
21 glioblastomas
5 anaplastic astrocytoma
5 anaplastic oligodendroglioma
The statistical significance would be much reduced because the sample size is also reduced.
However, the description of the relationship between mutations and the type of high-grade glioma can be seen in supplementary material 1 and there are no significant differences that allow us to select genes.
- The statistical analyses and criteria for identifying genes of interest are not clearly described. More details are needed on the bioinformatics and statistical methods used.
The statistical and bioinformatic criteria for selecting the genes are detailed in the methodology section:
Mutated genes identification related with temozolomide resistance mechanisms.
The mutated genes identification related with temozolomide resistance mechanisms was carried out using the following procedures: A univariate analysis was per-formed, in which summary measures, including measures of central tendency, were calculated for quantitative variables, along with their respective measures of dispersion, according to the distribution of the variable (Shapiro-Wilk test). For qualitative variables, absolute and relative frequencies were calculated. The results were presented through graphs and tables. To perform the bivariate analysis, the Logrank test, simple Cox regression models, Kaplan-Meier analysis, and survival curves were used. The results were presented in tables and a survival curve graph. To perform the multivariate model, a multiple Cox regression model was performed. For the entry of the variables into the final model, the behavior of the bivariate analysis was considered; that is, those variables that met the following criteria were entered into the model: statistical significance with p< 0.05, Hosmer Lemechow criterion with p< 0.25, and according to the criteria of the investigator (biological plausibility). Variables were entered progressively, and interaction analysis was performed for each model until reaching the final model, considering the principle of parsimony. Statistical analyses were performed using STATA (version 14) and SPSS (version 28) statistical programs.
- The functional significance of the identified mutations is not well characterized. Additional experiments would be needed to demonstrate a causal role in temozolomide resistance.
A thorough bioinformatics study of the effect of the mutations found in the four proteins analyzed is carried out. See the new results and analysis in table 3.
The new bioinformatics programs used are:
- Polyphen2
-SIFT
-PhD SNP
-SNP & GO
-MUpro
-I-Mutant3.0
-nsSNP
-TM-Aling
-SWISS-MODEL
-Project Hope
-Mutation 3D
These programs complement the analysis previously performed
- The discussion section is overly long and speculative given the limited data presented. This should be considerably shortened and focused on the actual findings.
The discussion was shortened in length and limited to the research findings.
- The English writing needs substantial improvement throughout. There are numerous grammatical errors and awkward phrasings that make the manuscript difficult to follow in places.
The language was corrected throughout the text.
Specific comments:
- The introduction lacks a clear rationale for focusing on these particular 4 genes. Why were these selected for analysis out of the many gene sequenced?
The evaluator is right, the following paragraph is added in the introductory part, in which the information available on the 4 genes is explored in depth.
In our study, we sequenced 324 cancer-related genes in 31 patients with high-grade gliomas. We found that mutations in the genes PIK3C2B, ERBB3, KIT, and MLH1 were related with resistance to temozolomide treatment. For this reason, the study focused on these genes. The etiology of glioma is still unknown (12), however several genes that affect the cell cycle and DNA repair have been proposed that play a role in the pathogenesis and progression of glioma. At present, the process of gliomagenesis has not been fully elu-cidated, however, some genes that intervene in the process have been identified. For example, the KIT gene, which encodes a class III receptor tyrosine kinase (RTK) (13), is frequently involved in tumorigenic processes. In the case of gliomas, it has been iden-tified that genesis and progression can be driven in part by populations of cancer stem cells, which have self-renewal and differentiation potential (14). It is known that some gliomas can arise from a cell with properties like those of neural stem cells (15). In this case, KIT is expressed in human bone marrow progenitor cells and in rodent glial pro-genitor cells, but its expression is lost when glial cells differentiate into oligodendro-cytes (16). The stem cell factor is upregulated in high-grade human gliomas, promotes angiogenesis and its expression is associated with a short survival. Stem cell factor and KIT activation may play a role in the development and progression of glioma (13), and its mutation frequency is 4.4% in glioblastomas (17).
On the other hand, the PIK3C2B gene is located, a member of the PIK3 family, which are lipid kinases involved in multiple cellular processes, including proliferation, differentiation, metabolism and survival (18). This gene is involved in the PI3K/Akt/mTOR signaling pathway, which has been confirmed to play an important role in the genesis and development of glioma (19). Glioblastoma patients with an ac-tivated PI3K/Akt/mTOR pathway also have a worse prognosis than patients without oncogenic activation of the pathway (19). PIK3Ks transduce signals from various growth factors and cytokines into intracellular messages by generating phospholipids, which in turn activate the serine/threonine kinase AKT and other downstream signaling path-ways, within these pathways is the tumor suppressor gene PTEN, which is the most important negative regulator of the PI3K/Akt/mTOR signaling pathway (20). The mu-tation frequency of PIK3C2B in high-grade gliomas is unknown.
Mutations in proteins involved in DNA mismatch repair can lead to immuno-histochemical losses of protein expression such as MSH2, MSH6, MLH1 leading to a hyper-mutable phenotype that can correlate with anti-PD1 efficacyb (21), which leads to the generation of neoantigens that can activate the immune system and promote anti-tumor activity, in fact, Hodges et al demonstrated that immunohistochemical loss of at least one protein involved in the mismatch repair pathway was associated with the hypermutation profile in patients with glioma (22). Additionally, hypermutation gen-otype or a high tumor mutational burden can also be induced by tomozolomide treat-ment (23). It is known that recurrent gliomas, especially those that are IDH mutants, can acquire mutations in DNA repair pathway genes and therefore develop resistance to chemotherapy (23), the mutation frequency of the MLH1 gene is 14% in high-grade gliomas (24).
On the other hand, the ERBB3 gene which belongs to the epidermal growth factor receptor (EGFR) family, which encompasses 4 members that are activated upon lig-and-induced homodimerization or heterodimerization (25), this gene is associated with the regulation of normal cell proliferation, apoptosis, differentiation and survival (26). When phosphorylated, ERBB3 shows a peculiar ability to maintain the pro-survival signaling of phosphatidylinositol 3-kinase (PI3K)/AKT, which may contribute to re-sistance against conventional and targeted therapies, and recurrence in a wide range of tumors. Therefore, ERBB3 itself is the subject of increasing interest as a therapeutic target (25). Overexpression of ERBB3 has been shown to be associated with short sur-vival of primary glioblastoma and is more frequent (up to 62%) in recurrent gliomas (25), suggesting that high ERBB3 activity may be involved with tumor aggressiveness and resistance to chemotherapy. Overexpression and activation of this gene sustains en-hanced cellular metabolism, with increased extracellular acidification, oxidative phos-phorylation and de novo fatty acid biosynthesis, which correlates with increased sur-vival in vitro and tumorigenic potential in vivo (25).
- Figure 4 is of poor quality and difficult to interpret. This should be redrawn for clarity.
The quality of the figure was improved, and its legend was better described.
- The bioinformatics analyses in section 3.6 are not well integrated with the clinical data. How do these predicted functional effects relate to the observed clinical outcomes?
The research aims to link mutations in the genes PIK3C2B, ERBB3, KIT, and MLH1 with resistance to temozolomide in patients with high-grade gliomas. The research did not aim to link the mutations with clinical outcomes.
- References need to be formatted consistently according to journal guidelines.
References were formatted according to Biomedicines journal standards.
Comments on the Quality of English Language
The manuscript contains numerous grammatical errors that significantly impair readability. Here are some examples of issues that need to be addressed:
Subject-verb agreement error: "Its instability index is computed to be 46.22 and classified as unstable."
This sentence was corrected
->"Its instability index is computed to be 46.22 and it is classified as unstable."
This sentence was corrected
Incorrect preposition use: "No signal peptide was detected using the SignalP bioinformatics program (version 6.0)."
This sentence was corrected
->"No signal peptide was detected by the SignalP bioinformatics program (version 6.0)."
This sentence was corrected
Additionally, the language was corrected throughout the text.
¡¡¡We greatly appreciate the evaluator's corrections, which help to greatly improve the research article!!!
Round 2
Reviewer 1 Report
Comments and Suggestions for Authors
The manuscript was heavily revised; however, several points still require attention and correction. The major point regarding WHO CNS Tumour classification arose. Line 67 states “grade IV or glioblastoma multiforme” which is not the 2021 WHO CNS Tumour classification, as Grade IV tumours were divided into glioblastomas and grade 4 astrocytomas (term glioblastoma multiforme was obsolete). Please refer to the current edition of the Blue Book (or relevant papers such as doi: 10.1016/j.clinme.2024.100240) and correct the section. Moreover, as stated in lines 154-155, it is expected that all cases in the work were classified according to current CNS tumours classification. However, as it can be seen from Supplementary Materials (which was unfortunately missed in the previous version), the outdated classification was used in the work. For example, patient 13 should be classified as Grade 4 Astrocytoma. In the table, some cases were reclassified, but not all. Also, case 16 (and some others) represents an odd situation – IDH is marked as mutated, but mutation in this gene was not found. Please, discuss it in the main text, and discuss a reclassification and mark the actual classification (according to 2021 classification, removing obsolete entities such as oligoastrocytoma) in the “glioma reclassification” column for all cases, marking significantly changing diagnosis. Then, make the changes in the main text and figures (Figure 1 should be redesigned) throughout the manuscript to address the altered diagnoses. It is stated (lines 746-748) that cases were reclassified, however the Results section uses outdated classification, while the Discussion section uses mixed classification, which is definitely inappropriate. If the authors want to compare survival or other outcomes between 2016 and 2021 classifications – it should be stated clearly, as I am not sure what the authors mean in a hard-to-read paragraph at lines 753-774.
The section in lines 86-139 does not seem to be introductory. Consider placing it in the Discussion section with simultaneous reduction throughout the Discussion to make it more concise.
Figure 3. The first column is marked as two cycles, but the figure caption mentions one cycle. Please correct.
Figure 4. Consider altering the figure legends from “kit = 0 kit = 1” to “kit-negative kit-positive”, etc.
Lines 894-895. The reference to such multiple sources (48-64) needs to be more concrete. Please make the citation more specific in the text or remove redundancy.
Please, make a summary at the conclusion section more detailed.
Author Response
Comments and Suggestions for Authors
The manuscript was heavily revised; however, several points still require attention and correction.
The major point regarding WHO CNS Tumour classification arose. Line 67 states “grade IV or glioblastoma multiforme” which is not the 2021 WHO CNS Tumour classification, as Grade IV tumours were divided into glioblastomas and grade 4 astrocytomas (term glioblastoma multiforme was obsolete). Please refer to the current edition of the Blue Book (or relevant papers such as doi: 10.1016/j.clinme.2024.100240) and correct the section.
The reviewer is right, the paragraph was modified as follows:
Tumors originating from glial cells are called gliomas. Glial cells are composed of four cell types: astrocytes, oligodendrocytes, ependymal cells, and microglia. The fifth edition of the World Health Organization's classification of central nervous system tumors published in 2021 indicates that the classification of these tumors has changed, leaving behind terms such as grade 4 glioblastomas multiforme, oligoastrocytoma, and anaplastic astrocytoma (6). The new classification recognizes 22 new tumor types and emphasizes the importance of histological and molecular diagnoses (7). In the case of diffuse gliomas in adults, there are astrocytoma, which range from grade 2 to 4. It has been recognized that grade 4 IDH-mutated astrocytoma is described as a biologically distinct entity from glioblastoma (6). Oligodendrogliomas are defined as gliomas with IDH mutation and 1p/19q codeletion, and they also usually have a mutation in the TERT promoter. Additionally, they are classified as grade 2 or 3 central nervous system tumors according to the WHO, depending on proliferation and anaplasia (8). The most common IDH wild-type glioma is glioblastoma, grade 4. These tumors have an astrocytic morphology and show high-grade morphological features including necrosis and/or microvascular proliferation (6). However, recent studies have demonstrated the existence of cellular heterogeneity within tumors, finding dynamic cell populations that may be related to patient survival (9).
Moreover, as stated in lines 154-155, it is expected that all cases in the work were classified according to current CNS tumours classification.
The reviewer is right, therefore:
The 2016 classification of central nervous system tumors was removed from the entire document and only the 2021 classification of central nervous system tumors remains.
However, as it can be seen from Supplementary Materials (which was unfortunately missed in the previous version), the outdated classification was used in the work. For example, patient 13 should be classified as Grade 4 Astrocytoma. In the table, some cases were reclassified, but not all. Also, case 16 (and some others) represents an odd situation – IDH is marked as mutated, but mutation in this gene was not found.
The obsolete classification of the supplementary material was removed and classification errors were corrected.
Please, discuss it in the main text, and discuss a reclassification and mark the actual classification (according to 2021 classification, removing obsolete entities such as oligoastrocytoma) in the “glioma reclassification” column for all cases, marking significantly changing diagnosis.
The reviewer is right, therefore:
The 2016 classification of central nervous system tumors was removed from supplementary material and only the 2021 classification of central nervous system tumors remains.
Then, make the changes in the main text and figures (Figure 1 should be redesigned) throughout the manuscript to address the altered diagnoses.
The figure was modified according to the new classification of central nervous system tumors (2021).
It is stated (lines 746-748) that cases were reclassified, however the Results section uses outdated classification, while the Discussion section uses mixed classification, which is definitely inappropriate.
Only the new classification of central nervous system tumors is included throughout the document and the supplementary material.
If the authors want to compare survival or other outcomes between 2016 and 2021 classifications – it should be stated clearly, as I am not sure what the authors mean in a hard-to-read paragraph at lines 753-774.
The comparison between the results of the classification between 2016 and 2021 was eliminated, leaving only the 2021 classification, because the current classification is more precise and does not use obsolete terms.
The section in lines 86-139 does not seem to be introductory. Consider placing it in the Discussion section with simultaneous reduction throughout the Discussion to make it more concise.
This paragraph was included in the introduction at the suggestion of reviewer 2. But you are right, it will be included in the discussion part.
Figure 3. The first column is marked as two cycles, but the figure caption mentions one cycle. Please correct.
The evaluator is right. The first relapse occurred in the second cycle of treatment with temozolomide. The explanation in Figure 3 was corrected.
Figure 3. The figure shows the number of treatment cycles with metronomic temozolomide that patients received after the first relapse and until the second relapse occurs. The largest number of patients presented a second relapse at the sixth treatment cycle (10 patients), but the results were highly variable, with early (second cycle) and late (21 cycles) second relapses.
Figure 4. Consider altering the figure legends from “kit = 0 kit = 1” to “kit-negative kit-positive”, etc.
The figure was edited, placing the red lines with the mutated genes and the blue lines with the genes without mutations. Additionally, the legend of the figure was modified.
Lines 894-895. The reference to such multiple sources (48-64) needs to be more concrete. Please make the citation more specific in the text or remove redundancy.
A more specific citation was made, and the redundancy was removed.
The paragraph read as follows:
According to the previous biological functions, three mutated proteins could be related to the gliomagenesis process and one of them to the mechanisms of resistance to temozolomide. Regarding to gliomagenesis process; the KIT gene, which encodes a class III receptor tyrosine kinase (RTK) (12, 13), is frequently involved in tumorigenic processes. In the case of gliomas, it has been identified that genesis and progression can be driven in part by populations of cancer stem cells, which have self-renewal and differentiation potential (14). It is known that some gliomas can arise from a cell with properties like those of neural stem cells (15). In this case, KIT is expressed in human bone marrow progenitor cells and in rodent glial progenitor cells, but its expression is lost when glial cells differentiate into oligodendrocytes (16). The stem cell factor is upregulated in high-grade human gliomas, promotes angiogenesis and its expression is associated with a short survival. Stem cell factor and KIT activation may play a role in the development and progression of glioma (13), and its mutation frequency is 4.4% in glioblastomas (17).
On the other hand, the PIK3C2B gene is located, a member of the PIK3 family, which are lipid kinases involved in multiple cellular processes, including proliferation, differentiation, metabolism and survival (18). This gene is involved in the PI3K/Akt/mTOR signaling pathway, which has been confirmed to play an important role in the genesis and development of glioma (19). Glioblastoma patients with an activated PI3K/Akt/mTOR pathway also have a worse prognosis than patients without oncogenic activation of the pathway (19). PIK3Ks transduce signals from various growth factors and cytokines into intracellular messages by generating phospholipids, which in turn activate the serine/threonine kinase AKT and other downstream signaling pathways, within these pathways is the tumor suppressor gene PTEN, which is the most important negative regulator of the PI3K/Akt/mTOR signaling pathway (20). The mutation frequency of PIK3C2B in high-grade gliomas is unknown. On the other hand, the ERBB3 gene which belongs to the epidermal growth factor receptor (EGFR) family, which encompasses 4 members that are activated upon ligand-induced homodimerization or heterodimerization (25), this gene is associated with the regulation of normal cell proliferation, apoptosis, differentiation and survival (26). When phosphorylated, ERBB3 shows a peculiar ability to maintain the pro-survival signaling of phosphatidylinositol 3-kinase (PI3K)/AKT, which may contribute to resistance against conventional and targeted therapies, and recurrence in a wide range of tumors. Therefore, ERBB3 itself is the subject of increasing interest as a therapeutic target (25). Overexpression of ERBB3 has been shown to be associated with short survival of primary glioblastoma and is more frequent (up to 62%) in recurrent gliomas (25), suggesting that high ERBB3 activity may be involved with tumor aggressiveness and resistance to chemotherapy. Overexpression and activation of this gene sustains enhanced cellular metabolism, with increased extracellular acidification, oxidative phosphorylation and de novo fatty acid biosynthesis, which correlates with increased survival in vitro and tumorigenic potential in vivo (25).
Regarding to resistance mechanism, alterations in genes that are part of the DNA mismatch repair system have been reported in recurrent glioblastomas after the use of temozolomide (65). Analysis of the Cancer Genome Atlas (TCGA) revealed a hypermutator phenotype, with mutations in at least one of the MMR genes (MLH1, MSH2, MSH6, or PMS2), suggesting either an escape from MGMT methylation or the selection of MMR mutated clones (66). Felsberg et al. reported changes in promoter methylation and expression of the MGMT, MLH1, MSH2, MSH6, and PMS2 genes after relapse in 80 patients with glioblastomas. They found that only four patients (6.25%) had a loss or decreased methylation of the MGMT promoter at recurrence. Although none of the four genes that are part of MMR had promoter hypermethylation, they did have mutations, which was confirmed by IHC (67). DNA double-strand breaks by temozolomide activate the homologous and non-homologous repair mechanisms, which, through the ataxia telangiectasia (ATM)-checkpoint kinase 2 (CHK2) pathways, lead to p53 damage repair caused by temozolomide or trigger apoptosis (68). Other mechanisms of resistance to temozolomide consist of the proper functioning or overactivation of some of the BER components, such as APGN, which would repair methylation at N7 of guanine and N3 of adenine (69). Resistance to temozolomide can develop due to the overactivation of MDM2, which increases the X-linked inhibitor of the apoptosis protein (XIAP). This key regulator of both intrinsic and extrinsic programmed cell death signaling functions by suppressing the activation of caspases 3, 7, and 9, triggering their degradation mediated by ubiquitination or by the improper functioning of p53; therefore, it is unable to activate the Bcl2 family and the activation of the DR5 receptor (70). Therefore, the MLH1 mutated gene could be involved in one of these resistance mechanisms.
Please, make a summary at the conclusion section more detailed.
The conclusion was written in more detail:
Conclusions: considering the early relapses of patients before 6 cycles of treatment with temozolomide, the decreased survival in patients with the four mutated genes, the structural alterations in the four proteins, the functional alterations caused by the mutations affecting the functional domains and decreasing the stability of the four mutated proteins, and the possible loss of interactions and interruption of metabolic pathways important for proper cellular functioning. We suggest that mutations in the four genes, and methylation of the gene promoter that codes for the MGMT protein could be related to response to temozolomide treatment.
¡¡¡Thank you very much for taking the time to read the research and making wonderful observations that helped improve the quality of the manuscript!!!
Reviewer 2 Report
Comments and Suggestions for Authors
The authors have made efforts to address the previous concerns, but several critical issues remain that need to be addressed for the manuscript to be suitable for publication.
1. While you have acknowledged the small sample size as a limitation, the conclusions in the manuscript still appear too definitive; could you revise the conclusions to reflect the exploratory and preliminary nature of your findings given the limited statistical power?
2. The reliance on bioinformatics analyses without experimental validation means the functional significance of the identified mutations remains speculative; can you discuss the limitations of in-silico predictions and outline any plans for future experimental validation to support your conclusions?
3. Combining different types of high-grade gliomas in your analysis may introduce confounding factors due to tumor heterogeneity; can you provide stratified analyses for each tumor type or explain how your statistical models account for this heterogeneity to strengthen the validity of your results?
Author Response
Comments and Suggestions for Authors
The authors have made efforts to address the previous concerns, but several critical issues remain that need to be addressed for the manuscript to be suitable for publication.
- While you have acknowledged the small sample size as a limitation, the conclusions in the manuscript still appear too definitive; could you revise the conclusions to reflect the exploratory and preliminary nature of your findings given the limited statistical power?
The evaluator is right, the conclusions were modified as follows.
Conclusions: Conclusions: considering the early relapses of patients before 6 cycles of treatment with temozolomide, the decreased survival in patients with the four mutated genes, the structural alterations in the four proteins, the functional alterations caused by the mutations affecting the functional domains and decreasing the stability of the four mutated proteins, and the possible loss of interactions and interruption of metabolic pathways important for proper cellular functioning. We suggest that mutations in the four genes, and methylation of the gene promoter that codes for the MGMT protein could be related to response to temozolomide treatment. These results are exploratory and preliminary but will be the basis for a subsequent experimental study to verify the validity of these findings.
- The reliance on bioinformatics analyses without experimental validation means the functional significance of the identified mutations remains speculative; can you discuss the limitations of in-silico predictions and outline any plans for future experimental validation to support your conclusions?
The author is right, so we add in the following paragraph at the end of the discussion admitting that future studies should be carried out with a larger number of samples and verify the bioinformatics findings with experimental verifications. However, this research does not have the money to carry out an experimental verification.
paragraph added:
Bioinformatics studies are of great importance because they allow us to organize information from large volumes of scientific data, analyze them, and obtain preliminary conclusions about the results of the research. These results can be verified experimentally to corroborate their validity; however, having bioinformatics results directs experimental studies to obtain better results. The sample size of 31 patients is very small for a genomic study. This limits the statistical power and generalizability of the findings. Therefore, we recognize that the sample size is the main limitation of the study.
- Combining different types of high-grade gliomas in your analysis may introduce confounding factors due to tumor heterogeneity; can you provide stratified analyses for each tumor type or explain how your statistical models account for this heterogeneity to strengthen the validity of your results?
Regarding this topic, we would like to tell you about the genesis of the project:
- Patients with High-grade gliomas in the city of Medellin are few and difficult to recruit.
- During the two years of the project, only 31 patients were recruited.
- The 31 patients had different types of high-grade gliomas.
- When performing the stratified analysis by type of high-grade glioma, no significant differences were found, because the sample size is greatly reduced for each type of glioma.
- For this reason, the analysis was carried out in a general way to see how mutations influence resistance to temozolomide at a general level in the different types of High-grade gliomas.
- For these reasons we present the results in this way and it does not make logical sense to perform a stratified analysis, since the sample size is reduced in each type of High-grade gliomas and significant differences are not found when performing the analysis for each type of high-grade glioma.
This is a preliminary study, and it is hoped that future studies will be able to collect a larger number of patients and perform stratified analyses.
¡¡¡Thank you very much for taking the time to read the research and making wonderful observations that helped improve the quality of the manuscript!!!
Round 3
Reviewer 1 Report
Comments and Suggestions for Authors
No other issues, the manuscript sounds good.